# Chronic neuronal activation increases dynamic microtubules to enhance functional axon regeneration after dorsal root crush injury

Di Wu[1], Ying Jin[1], Tatiana M. Shapiro[1], Abhishek Hinduja[1], Peter W. Baas [1] & Veronica J. Tom [1✉]

After a dorsal root crush injury, centrally-projecting sensory axons fail to regenerate across the dorsal root entry zone (DREZ) to extend into the spinal cord. We find that chemogenetic activation of adult dorsal root ganglion (DRG) neurons improves axon growth on an in vitro model of the inhibitory environment after injury. Moreover, repeated bouts of daily chemogenetic activation of adult DRG neurons for 12 weeks post-crush in vivo enhances axon regeneration across a chondroitinase-digested DREZ into spinal gray matter, where the regenerating axons form functional synapses and mediate behavioral recovery in a sensorimotor task. Neuronal activation-mediated axon extension is dependent upon changes in the status of tubulin post-translational modifications indicative of highly dynamic microtubules (as opposed to stable microtubules) within the distal axon, illuminating a novel mechanism underlying stimulation-mediated axon growth. We have identified an effective combinatory strategy to promote functionally-relevant axon regeneration of adult neurons into the CNS after injury.

---

[1] Department of Neurobiology and Anatomy, Marion Murray Spinal Cord Research Center, Drexel University College of Medicine, Philadelphia, PA, USA.
✉email: vjt25@drexel.edu

Unlike the peripherally-projecting axon of dorsal root ganglia (DRG) neurons, the central branch cannot successfully regenerate after trauma – such as to the dorsal columns tract within the spinal cord or a dorsal root—and often results in permanent sensory deficits[1,2]. Interestingly, after a dorsal root injury, the centrally projecting axons attempt to regenerate as long as the root is contiguous, allowing for the alignment of Schwann cells upon which these axons extend. However, this advancement ceases when the growing axon tip reaches the dorsal root entry zone (DREZ), the interface between the peripheral nervous system (PNS) and the central nervous system (CNS). The failure of sensory axon regeneration after dorsal root injury is partly attributed to a CNS environment that is less favorable for growth[3,4]. After injury, the CNS is rich in growth inhibitory proteins including Nogo, myelin-associated glycoprotein, and chondroitin sulfate proteoglycans (CSPG) and lacks trophic support[4–11]. Thus, at the DREZ, the transition from

**d**

**Axon crossing of inhibitory rim**

a permissive to a hostile environment stalls growth cone advancement[5,12,13]. We and others have administered the bacterial enzyme chondroitinase ABC (ChABC) to digest upregulated CSPGs after dorsal root crush[14,15]. This allows for a limited number of axons to regenerate across the DREZ, indicating that modifying the inhibitory environment alone is not sufficient to obtain robust axon regeneration.

A limited intrinsic regenerative capacity of adult sensory axons also contributes to the failure of axon regeneration[16,17]. Lesions of DRGs' central axon branches fail to elicit a robust regenerative response[18,19]. Some interventions have been shown to enhance the intrinsic regenerative response of sensory neurons, including a so-called peripheral "conditioning lesion"[20–22], induction of inflammation near the cell body within the ganglia themselves[14,23], and driving activation of mTOR in DRG neurons[15]. More recently, we found that increasing the labile microtubule mass in adult DRG neurons by knocking down the microtubule-severing protein fidgetin improves axon regeneration across the DREZ after dorsal root crush[24].

Another avenue that has been explored to increase the intrinsic capacity for axon regeneration is neuronal activation. Electrical stimulation can enhance regeneration of sensory axons after peripheral nerve or dorsal columns injury[25,26] and sprouting of cortical axons into contralateral spinal cord gray matter after pyramidotomy[27,28]. Moreover, the growth promoting effect of neuronal stimulation is not limited to electrical stimulation. Remotely activating neurons expressing the excitatory designer receptor exclusively activated by designer drugs (DREADDs) hM3Dq using the receptor's ligand clozpine-N-oxide (CNO)[29–31] enhances retinal ganglion cell axon growth after an optic nerve crush[32,33].

The use of chemogenetics to activate neurons is particularly intriguing because specificity of activation can be bestowed by regulating which neurons express hM3Dq. Moreover, hM3Dq$^+$ neurons can be activated over chronic periods of time relatively easily using systemic injections of CNO, which can cross the blood-brain barrier. Because neuronal activity plays a crucial role in the normal development and refinement of circuits, it is possible that the neuronal activation would need to be delivered over a prolonged period of time in order to not only promote axon regeneration but to also enhance the functional connectivity of the axons that do regenerate.

In the current study, we hypothesized that combining long-term, repeated (i.e., daily) neuronal activation of adult DRG neurons (using chemogenetics) after dorsal root crush with ChABC modulation of the inhibitory environment at the DREZ would enhance functional sensory axon regeneration into spinal cord.

**Fig. 1 Chemogenetic DRG neuron activation promotes neurite crossing of a ChABC-treated inhibitory proteoglycan barrier.** Representative images of DRG cultures growing on ChABC-treated aggrecan spots are shown in **a**–**c**. The inhibitory rim is demarcated in **a**. Quantification of the numbers of axons crossing the inhibitory rim is shown in **d**. Neurons and axons were visualized by βIII-tubulin staining (green). Without ChABC, neurons transduced with either AAV-mCherry or -hM3Dq failed to traverse the inhibitory rim. ChABC alone significantly enhanced axonal growth from both mCherry$^+$ DRGs in the presence of CNO or hM3Dq$^+$ DRGs in the absence of CNO. After ChABC digestion of CSPG, some axons from control, mCherry$^+$ DRG neurons were able to grow across the inhibitory rim (**a**, arrowhead), similar to what we observed when hM3Dq$^+$ DRG neurons are cultured in the absence of CNO (**b**, **d**). CNO-mediated activation of hM3Dq$^+$ DRG neurons enabled more neurites to cross the inhibitory rim (arrows in **c**, **d**). More axons crossed the inhibitory rim when CNO-mediated chemogenetic activation of hM3Dq$^+$ DRGs was combined with ChABC digestion of the inhibitory substrate. $N = 16$ spots/group. Mean ± SEM. One-way ANOVA and post-hoc multiple comparisons testing using the two-stage step-up method of Benjamini, Krieger, and Yekutieli, **$p = 0.0054$, ***$p < 0.001$ (hM3Dq+CNO-ChABC vs. mCherry+CNO + ChABC $p = 0.0020$; mCherry+CNO + ChABC vs. hM3Dq+CNO + ChABC $p = 0.0003$), ****$p < 0.0001$. Scale bar: 50 μm. Source data are provided as a Source Data file.

## Results

**CNO-induced activation of adult, hM3Dq$^+$ DRGs enhances neurite growth on CSPG spots.** We first sought to determine if activation of adult DRG neurons could enhance axonal regrowth in an in vitro model that mimics the gradient of upregulated CSPGs present at the DREZ after a dorsal root crush[14,15,34]. Adult, dissociated DRG neurons were transduced with AAV5-hSyn-hM3Dq-mCherry or AAV5-hSyn-mCherry to express hM3Dq/mCherry or mCherry only, respectively. Approximately 90% of βIII-tubulin$^+$ neurons strongly expressed the fluorescent reporter (mCherry: 88.59 ± 2.71%; hM3Dq: 88.22 ± 2.59%). No difference in transduction efficiency was observed between groups.

The transduced neurons were then plated onto coverslips with spots of a mixture of the inhibitory CSPG aggrecan and the growth promoting molecule laminin. Normally in this assay,

DRG neurons are able to attach in the center of each CSPG spot, where the concentration of CSPG is low. Similar to what we saw before[14,15], their axons were not able to extend across the rim of the spot, where there is a high concentration of CSPG, unless CSPGs in the substrate were digested with ChABC (Fig. 1a–c). In the absence of ChABC, $\beta$III-tubulin$^+$ neurites from hM3Dq$^+$ and mCherry$^+$ neurons were unable to cross the spot's rim (Fig. 1d). This suggests CNO-induced activation on hM3Dq$^+$ neurons alone was not sufficient to overcome the inhibition from high concentration of aggrecan.

When CSPGs in the substrate were digested with ChABC, only some axons from from mCherry$^+$ DRGs (Fig. 1a) or unactivated, hM3Dq$^+$ DRGs (i.e., cultures in the absence of CNO but in the presence of DMSO vehicle; Fig. 1b) crossed the rim, mimicking the improved regeneration of a percentage of axons after digesting the scar with ChABC. Importantly, when ChABC was combined with CNO-mediated activation of hM3Dq$^+$ DRGs, many more neurites (~65% more) traversed the rim (Fig. 1c, d; one-way ANOVA $F(4,15) = 22.39$, $p < 0.0001$; post-hoc hM3Dq$^+$ +CNO + ChABC vs.: mCherry$^+$+CNO-ChABC, $p < 0.0001$; hM3Dq$^+$+CNO-ChABC, $p < 0.0001$; mCherry$^+$+CNO + ChABC, $p = 0.0003$; hM3Dq$^+$+DMSO + ChABC, $p < 0.0001$). These data suggest that chemogenetically activating adult neurons allows more axons to overcome inhibition that remains after ChABC, resulting in even more growth.

**AAV-hSyn-hM3Dq primarily transduces large caliber DRGs in vivo.** We wanted to determine if the encouraging in vitro results translated to improved axon regeneration in vivo after injury. Since we used adult DRG neurons in the in vitro experiments, we turned to a dorsal root crush injury model to begin to determine if activating hM3Dq$^+$ DRG neurons increases functional axon regeneration into the CNS.

We first wanted to determine which DRGs were transduced in vivo. AAV5-hSyn-mCherry or AAV5-hSyn-hM3Dq-mCherry was carefully injected into the C6-C8 DRGs and mCherry expression was examined 4 weeks later. The vectors transduced DRG neurons with similar efficiency (~40% for both groups). Transduction was neuron specific, as all mCherry$^+$ cells were also $\beta$III-tubulin$^+$. Some of the transduced neurons were CGRP$^+$, small, peptidergic sensory neurons (24.1 ± 4.9%; Fig. 2a–c, j) or IB4$^+$, small, nonpeptidergic neurons (13.1 ± 1.7%; Fig. 2d–f, j), but the majority were NF-200$^+$, large diameter sensory neurons (62.3 ± 3.8%; Fig. 2g–j; one-way ANOVA $F(2,56) = 66.50$, $p < 0.0001$).

**Repeated activation of adult DRG neurons after dorsal root crush improves recovery in a sensorimotor task that requires proprioception.** ChABC-mediated digestion of glycosaminoglycan chains on CSPG can promote some axonal regeneration, including at the DREZ[35]. A single microinjection of ChABC into the dorsal horn effectively and widely digests upregulated CSPGs at the DREZ after dorsal root crush[15]. To determine if repeated activation of hM3Dq$^+$ DRGs further enhances regeneration across a ChABC-treated DREZ, AAV-hM3Dq or -mCherry was injected into the C6-C8 DRGs, as described above. One month later, when transgene expression in the neurons is high, the right C5-T1 dorsal roots were crushed to completely sever all of the axons within the root. This disrupts all sensory input from the distal, ipsilateral forepaw[36]. Because we found that neuronal activation alone (i.e., in the absence of ChABC) was insufficient to promote axon crossing on CSPG-containing inhibitory boundaries in our in vitro studies (Fig. 1d), we microinjected ChABC at the DREZ of all of the animals. All animals received daily, subcutaneous injections of CNO starting the day after the injury for

either 4 weeks or 12 weeks to account for any off-target effects CNO may have[37,38]. CNO administration did not robustly induce c-Fos expression, a well-established marker of neuronal activation[39–41], in mCherry$^+$ neurons (5.9 ± 1.3%; Fig. 3a–c, g) but did so in hM3Dq$^+$ neurons (55.3 ± 2.1%; Fig. 3d–g; $p < 0.0001$), confirming that subcutaneous CNO injections activates hM3Dq$^+$ neurons specifically. We did not notice any overt adverse effects of the chronic CNO injections. A summary of the in vivo paradigm is shown in Fig. 4.

Prior to the crush injury and weekly thereafter, we assessed sensory function of the mCherry$^+$ and hM3Dq$^+$ animals using the Hargreaves' test (thermal sensation) and the Von Frey filament test (fine touch). Because the majority of DRG neurons that are transduced by AAV5 are large caliber neurons (Fig. 2g–j), we also examined how well the animals were able to correctly place their ipsilateral forepaw while walking on a grid platform, a task that requires proprioception. All behavioral testing was done prior to that day's CNO administration to prevent the effects of acute neuronal activation from being a confound.

All animals had behavioral deficits acutely after the crush that are consistent with the complete interruption of afferent input from the distal, ipsilateral forepaw. In the grid walking test, both mCherry$^+$ and hM3Dq$^+$ animals had a comparable number of foot slips right after the injuries (Fig. 5a). mCherry$^+$ animals had a persistent deficit in this sensorimotor task across all time points. While hM3Dq$^+$ animals had a similar deficit at earlier post-injury time points, they correctly placed their ipsilateral paw more frequently starting 6 weeks post-injury for the duration of the study (Fig. 5a, two-way ANOVA $F(1,156) = 82.19$, $p < 0.0001$; post-hoc mCherry vs. hM3Dq at: 6 weeks, $p < 0.0001$; 7 weeks, $p = 0.0003$; 8 weeks, $p < 0.0001$; 9 weeks, $p = 0.0023$; 10 weeks, $p = 0.0049$; 11 weeks, $p = 0.0027$; 12 weeks, $p = 0.0065$). On the other hand, mCherry$^+$ and hM3Dq$^+$ animals had similar, pronounced, lasting deficits in both mechanical (Fig. 5b) and thermal sensation (Fig. 5c). These data support that repeated CNO-induced activation, particularly of large caliber, hM3Dq$^+$ DRG neurons, after complete dorsal root crush results in significant recovery in a behavioral task that is dependent upon sensorimotor integration.

**Recurrent chemogenetic activation of DRG neurons after dorsal root crush enhances axonal regeneration across the DREZ into spinal gray matter.** Does the improved sensorimotor function we observed in the hM3Dq$^+$ animals correlate with improved sensory axon regeneration into the spinal cord? We examined mCherry$^+$ axons that extended across the DREZ (as seen in Fig. 6a, b). As we and others found before, ChABC alone allowed for some axonal regeneration and we observed axons across the DREZ in both the mCherry$^+$ and hM3Dq$^+$ groups, including at 4 weeks post crush (Fig. 6e). Few of these axons extended into gray matter (Fig. 6f). There was no difference in the number of axons between groups at 4 weeks (two-way ANOVA $F(8, 72) = 0.5924$, $p = 0.781$).

At 12 weeks, we again observed mCherry$^+$ axons across the DREZ in both mCherry$^+$ and hM3Dq$^+$ animals (Fig. 6a, b, g). However, while neuronal activation did not increase the number of axons that grew across the DREZ (Fig. 6g; $p = 0.08$), it did impact where these axons extended. In the mCherry$^+$ animals, axons were found predominantly in white matter (arrows in Fig. 6a) and very few extended into gray matter (Fig. 6a, a′, h). However, in the hM3Dq$^+$ animals, we observed that more axons extended deeper into ipsilateral dorsal horn (Fig. 6b–d, h; two-way ANOVA $F(8,90) = 2.772$, $p = 0.0087$). These data indicate that chronic activation of DRG neurons increases axon regeneration into gray matter after dorsal root crush.

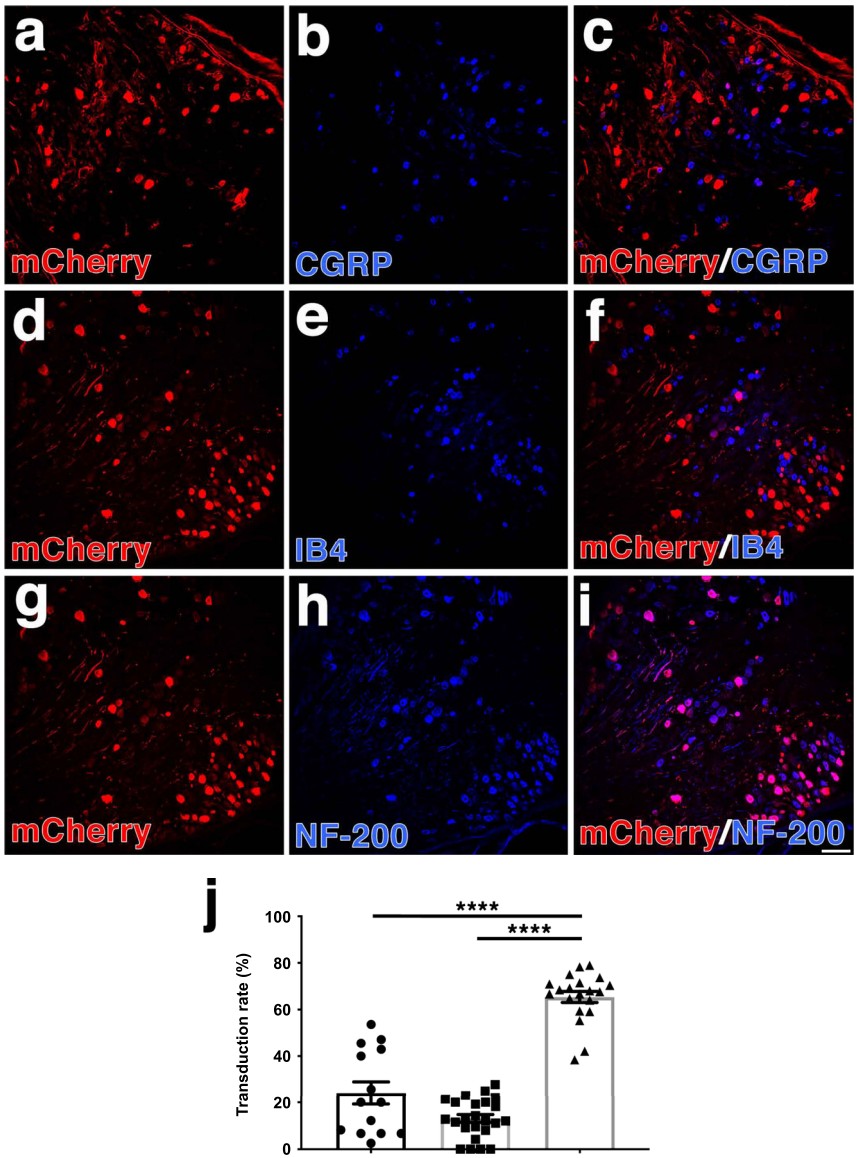

**Fig. 2 Intraganglionic injections of AAV5-hSyn-hM3Dq primarily transduces large caliber DRGs. a–i** AAV5-hSyn-hM3Dq was injected into the right C5-C8 DRGs. Four weeks later, the DRGs were isolated and immunostained for mCherry (**a**, **d**, **g**, red) and calcitonin gene related peptide (CGRP; **b**, blue), IB4 (**e**, blue) or NF-200 (**h**, blue). We observed mCherry expression mostly in large diameter, NF-200+ neurons (**g–i**) rather than nociceptive CGRP+ (**a–c**) or IB4+ neurons (**d–f**). **j** Quantification of the transduction rates in CGRP+, IB4+ and NF-200+ neurons demonstrates that AAV5 transduces NF-200+ neurons more efficiently than smaller caliber, nociceptive neurons. $N = 12$ animals. Mean ± SEM. One-way ANOVA and post-hoc multiple comparisons testing using the two-stage step-up method of Benjamini, Krieger, and Yekutieli, ****$p < 0.0001$. Scale bar: 100 μm. Source data are provided as a Source Data file.

**Axons that regenerate with neuronal activation form functional synapses upon spinal neurons.** Axon regeneration is not always followed by their integration into existing circuits or functional improvements[15,42,43]. We wanted to assess whether afferents that regenerated back into spinal cord upon repeated, CNO-induced activation of hM3Dq+ DRG neurons synapsed upon neurons in the dorsal horn. To do this, we electrically stimulated the ipsilateral ulnar and median nerves ipsilateral to the injury at 4 or 12 weeks after crush. The dorsal root crush injuries interrupted all of the sensory input from the median and ulnar nerves into the ipsilateral spinal cord. Thus, electrical stimulation of the ipsilateral median and ulnar nerves will only activate neurons within the spinal cord if they receive synaptic input from sensory axons that regenerate across the DREZ.

At 4 weeks after injury, when few axons regrew into gray matter regardless of group (Fig. 6f), there were few NeuN+ neurons that co-expressed c-Fos (such as those noted by arrowheads in Fig. 7a), an established marker of neuron activation, in both mCherry+ and hM3Dq+ animals (Fig. 7d). At 12 weeks after crush, when we observed that neuronal activation enhances axon regrowth into spinal gray matter (Fig. 6h), median and ulnar nerve stimulation induced c-Fos in significantly more cells in the hM3Dq+ animals than in mCherry+ animals (Supplemental Fig. 1a). This included c-Fos+ cells that were not NeuN+ (Supplemental Fig. 1b) that presumably are GFAP+ astrocytes (Supplemental Fig. 1c) as well as NeuN+ dorsal horn neurons (Fig. 7b, c, e; $p = 0.0008$). We found that the majority of the c-Fos+ neurons in the hM3Dq+ animals were located in laminae I-II (72.1 ± 0.7%) and the remaining were largely found in lamina III (27.3 ± 0.7%, Fig. 7f).

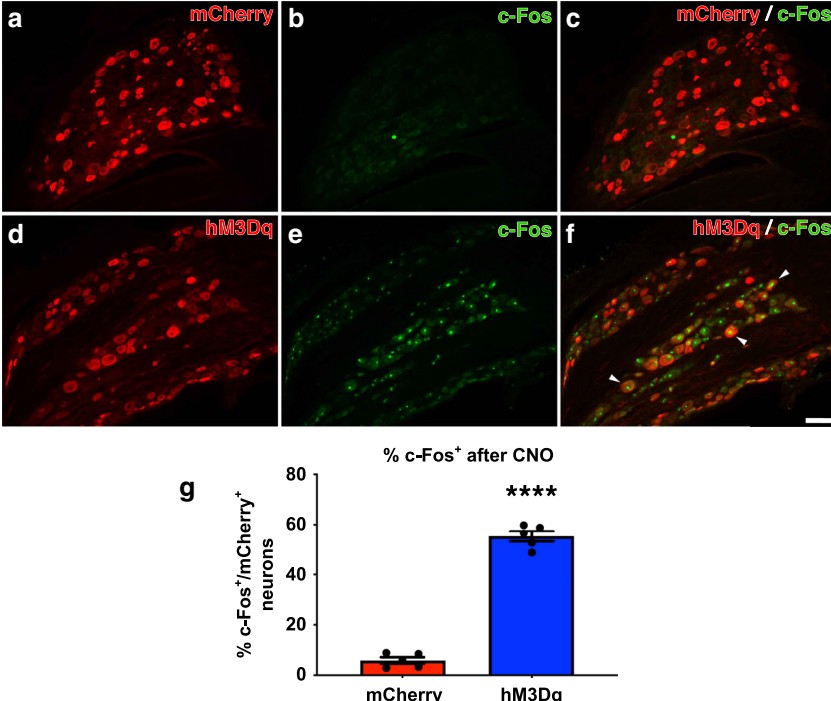

**Fig. 3 CNO administration activates adult, hM3Dq+ DRG neurons in vivo.** AAV5-hSyn-hM3Dq-mCherry or AAV5-hSyn-mCherry was injected into the right C5-C8 DRGs. Four weeks later, CNO was subcutaneously injected into all animals. Two hours later, animals were perfused and DRG sections were stained for the mCherry reporter and c-Fos, an established marker of neuronal activation (**a**–**f**). Very few neurons transduced with AAV-mCherry expressed c-Fos (**a**–**c**) while many hM3Dq+ neurons expressed c-Fos (**d**–**f**, arrowheads). **g** Quantification of the percent of mCherry+ neurons that are also c-Fos+ after CNO administration. These data confirm that subcutaneous CNO injection chemogenetically activated hM3Dq+ neurons in vivo. $N = 5$ animals per group. Mean ± SEM. Two-tailed unpaired $t$-test, ****$p < 0.0001$. Scale bar: 100 μm. Source data are provided as a Source Data file.

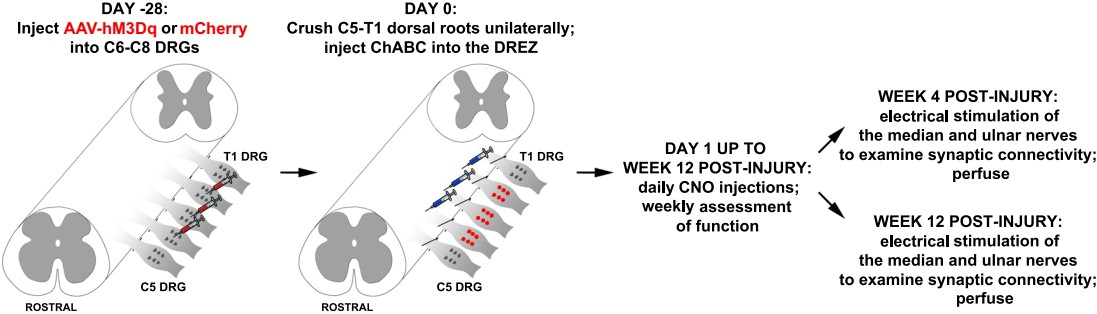

**Fig. 4 Timeline of the in vivo experiments.** Schematic summarizing the timeline and experimental procedures.

These data indicate that neuronal activation not only enhances axon regeneration into spinal gray matter but that these regenerated axons synapse upon interneurons and integrate into spinal circuits.

**Enhanced axonal outgrowth mediated by neuronal activation is dependent upon dynamic (i.e., labile) microtubules.** Microtubules are essential to the axon's differentiation, structural integrity, and growth[44,45]. Individual microtubules in the axon consist of two domains—a stable and a labile one—the latter of which is considerably more dynamic[46]. Microtubule stability is reflected in the extent of certain post-translational modifications of tubulin, namely acetylation and detyrosination[46]. Stable microtubules are hallmarked by being detyrosinated and acetylated while labile microtubules are tyrosinated and not acetylated, and this is the case of the labile and stable domain of axonal microtubules[46]. During normal development, the regulation of axon growth is thought to be mediated, at least in part, by

proteins that modulate the ratio of the stable (acetylated, detyrosinated) fraction and the labile (less acetylated, tyrosinated) fraction[47,48]. Indeed, rapidly growing axons tend to have a higher proportion of labile microtubule mass[49]. Moreover, we recently found that experimentally increasing the labile microtubule mass in adult DRG neurons enhances axon regeneration into the spinal cord after a dorsal root crush[24].

We wondered if neuronal activation increases axon regeneration by affecting the dynamic state of the microtubule array in the extending axon. We cultured adult mCherry+ or hM3Dq+ DRG neurons on poly-L-lysine in the presence of CNO. Twenty-four hours later, the cultures were stained for neuron specific ßIII-tubulin and either tyrosinated (i.e., labile) or acetylated (i.e., stable) tubulin. Because the distal end of the axon is important in determining axon regeneration[50], we focused our analysis on the distal-most 50 μm of the axon.

Extending axons from "unactivated" mCherry+ DRG neurons were full of ßIII-tubulin (Fig. 8a, a′). When we looked more

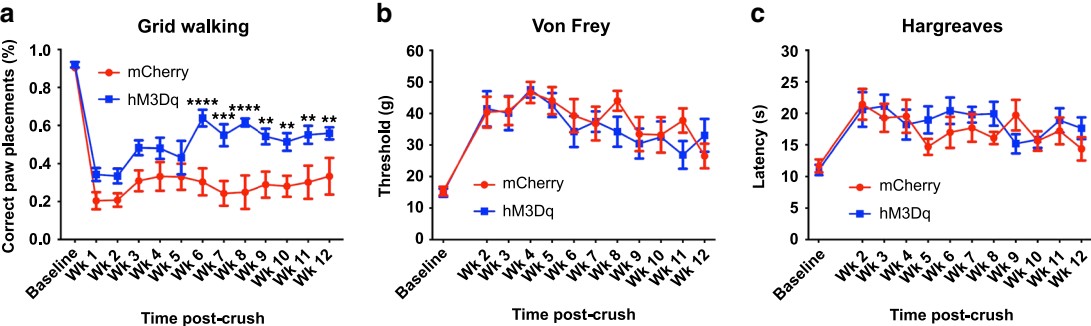

**Fig. 5 Repeated activation of adult hM3Dq+ DRG neurons improves proprioceptive functional recovery.** Sensory testing was performed before dorsal root crush (baseline testing) and every week after dorsal root crush from week 2 to week 12. **a** In the grid walking test, both animal groups had comparable deficits in their ability to correctly place their affected forepaw on rungs while walking on a gridded platform on the grid early on. While the deficit persisted in the mCherry+ animals, repeated, daily activation of hM3Dq+ DRGs with CNO resulted in improved paw placement starting 6 weeks post injury. **b** Dorsal root crush compromised mechanical sensation in the ipsilateral forepaw, as determined using Von Frey filaments. The response threshold did not differ between groups at any testing time point. **c** Thermal sensitivity was also diminished after dorsal root crush injuries, as indicated in the Hargreaves' test. Both animal groups showed increased response latency after the injury. There were no differences between groups. $N = 7$ animals per group. Mean ± SEM. Two-way ANOVA and post-hoc multiple comparisons testing using the two-stage step-up method of Benjamini, Krieger, and Yekutieli, $**p < 0.01$ (mCherry vs. hM3Dq Week 9 $p = 0.0023$, Week 10 $p = 0.0049$, Week 11 $p = 0.0027$, Week 12 $p = 0.0065$), $***p = 0.0003$, $****p < 0.0001$. Source data are provided as a Source Data file.

specifically at acetylated tubulin, a marker for stable microtubules, we found that the vast majority of the ßIII-tubulin+ axon shaft of mCherry+ neurons was also rich in acetylated tubulin. The only region that lacked appreciable immunoreactivity for acetylated tubulin was the very distal end of the axon, the presumptive growth cone (Fig. 8b, b'). Growing axons from CNO-activated, hM3Dq+ neurons had similar levels of immunofluorescence for ßIII-tubulin as mCherry+ axons (Fig. 8c, e; $p = 0.082$), indicating that neuronal activation did not affect total tubulin levels within the axon shaft. However, the hM3Dq+ axons had less acetylated tubulin within the distal portion of the ßIII-tubulin+ axon (Fig. 8d, d'), with a smaller ratio of acetylated tubulin to total tubulin (Fig. 8f, $p = 0.043$). These data indicate that neuronal activation decreases levels of acetylated microtubules at the end of growing axons.

To determine if the decrease in acetylated microtubules we observed in activated, hM3Dq+ neurons plays a role in their improved axon regrowth ability, we added the histone deacetylase inhibitor MS-275 to the culture media. MS-275 inhibited deacetylation of tubulin in the CNO-activated, hM3Dq+ neurons and increased the ratio of acetylated tubulin to total tubulin to levels comparable to that seen in growing axons from unactivated, mCherry+ neurons (Fig. 8g; one-way ANOVA $F(3,309) = 3.317$, $p = 0.020$; post-hoc hM3Dq+ vs. hM3Dq++MS-275, $p = 0.025$; hM3Dq++MS-275 vs. mCherry, $p = 0.772$).

Does increasing acetylated tubulin levels affect growth mediated by neuronal activation? To answer this, we returned to the in vitro CSPG spot assay we used in earlier experiments (Fig. 1). CNO-activated, hM3Dq+ neurons were better able to cross a ChABC-treated, inhibitory rim than control, mCherry+ neurons (Fig. 8h), replicating what we observed before. Increasing the levels of acetylated tubulin abolished this neuronal activation-mediated effect. In the presence of MS-275, CNO-activated, hM3Dq+ neurons extended fewer axons across the ChABC-treated rim than activated, hM3Dq+ neurons without MS-275 (one-way ANOVA $F(3,20) = 14.02$, $p < 0.0001$; post-hoc hM3Dq+ vs. hM3Dq++MS-275, $p = 0.0071$) and similar numbers of axons as mCherry+ controls (Fig. 8h; one-way ANOVA $F(3,20) = 14.02$, $p < 0.0001$; post-hoc mCherry+ vs. hM3Dq++MS-275, $p = 0.121$). These data demonstrate that neuronal activation mediates axon regrowth, at least in part, by decreasing acetylated tubulin, a marker of stable microtubules.

To obtain a more complete picture of the state of the microtubule array, we also assessed tyrosinated tubulin, a marker of labile microtubules. Unlike acetylated tubulin, tyrosinated tubulin was found throughout the distal-most portion of an extending, βIII-tubulin rich axon from mCherry+ neurons (Fig. 9a, b, b'). Interestingly, there was more tyrosinated tubulin in the distal axon of activated, hM3Dq+ DRG neurons (Fig. 9c, d, d', f; $p = 0.016$), indicating that neuronal activation increases tyrosinated microtubules at the end of actively extending axons. This increase in tyrosinated microtubules is independent of an increase in total tubulin, as neuronal activation did not increase total tubulin levels (Fig. 9e; $p = 0.082$), corroborating what we saw previously (Fig. 8e).

These data led us to ask if inhibiting detyrosination of microtubules—thus, increasing tyrosinated tubulin—is sufficient to enhance axon growth and adult neurons. To determine this, we used parthenolide, which inhibits tubulin carboxypeptidase to compromise the detyrosination of microtubules in DRG cultures[51]. Parthenolide treatment increased levels of tyrosinated tubulin in unactivated, mCherry+ DRGs (Fig. 9g; one-way ANOVA $F(3,300) = 6.294$, $p = 0.0004$; post-hoc mCherry+ vs. mCherry+ + parthenolide, $p < 0.0001$) to levels comparable to activated hM3Dq+ neurons (mCherry+ + parthenolide, vs. hM3Dq+, $p = 0.164$). Moreover, treating mCherry+ DRGs with parthenolide resulted in comparable crossing of the inhibitory rim in the CSPG spot assay as activated, hM3Dq+ DRGs (Fig. 9h; one-way ANOVA $F(3,12) = 3.817$, $p = 0.039$; post-hoc hM3Dq+ vs. mCherry+ + parthenolide, $p = 0.906$). These data indicate that increasing microtubule tyrosination in adult axons is sufficient to mimic the enhanced regrowth across normally inhibitory boundaries we saw with neuronal activation.

**Chronic chemogenetic activation increases mTOR activation in the DRG soma.** We and others previously found that mTOR activation increases sensory axon regeneration after injury[15,52]. We examined whether chemogenetic activation of DRGs here increases mTOR activation. One day after CNO administration, we stained sections of DRGs transduced with either mCherry or hM3Dq for active (i.e., phosphorylated) ribosomal protein S6 (p-S6), a downstream target of mTOR. While some mCherry+ neurons had detectable p-S6, many did not (Fig. 10a–c, arrowheads). However, the vast majority of hM3Dq+ DRGs had high

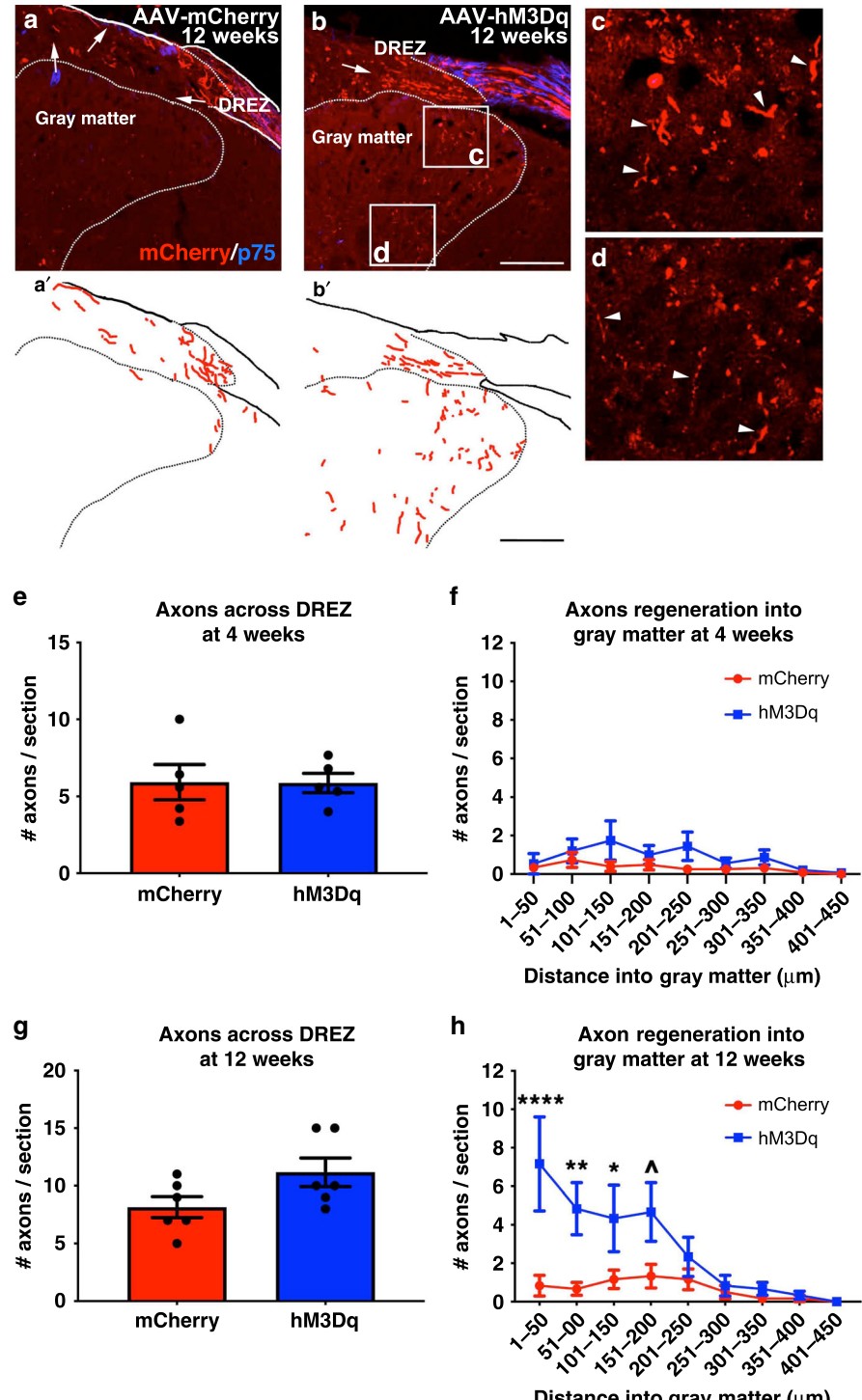

levels of p-S6 (Fig. 10d–f, arrows), even neurons that weakly expressed hM3Dq (asterisks). Overall, hM3Dq+ DRGs had more intense immunoreactivity for p-S6 (Fig. 10g; $p < 0.0001$), indicating that chemogenetic stimulation of adult DRGs activates mTOR.

## Discussion
Neuronal activity can profoundly influence functional restoration after both CNS and PNS injuries. Various forms of activity (e.g., electrical, optogenetic) can enhance sprouting of spared projections[27,53–55], regenerative axonal growth[19,56], remyelination[57,58], and synaptic formation[59,60]. Another means

to activate neurons in a relatively specific fashion is via chemogenetics. Previous studies have shown that chemogenetic activation of adult neurons increases the intrinsic capacity for axonal regeneration[32,33]. Because we and others have demonstrated that combining strategies to address both extrinsic and intrinsic limitations to axonal regeneration is more effective than solely addressing the extrinsic barriers[14,15,61,62], in the present study, we sought to determine if repeated chemogenetic activation would enhance DRG axon regrowth and, if so, by what mechanisms.

We found that chemogenetic activation of hM3Dq+ adult DRG neurons increases axon growth on a ChABC-treated inhibitory substrate in vitro (Fig. 1). Moreover, we found that

**Fig. 6 Recurrent, chemogenetic activation of hM3Dq$^+$ DRGs after a dorsal root crush promotes axon regrowth into the gray matter.** Transverse spinal cord tissue sections containing the dorsal root from animals 4 weeks or 12 weeks after dorsal root crush were immunostained for mCherry (red; to visualize axons from transduced DRG neurons) and p75 (blue; to visualize Schwann cells within the dorsal root to determine the boundary between the PNS and the CNS). Representative confocal images of the spinal cord sections with the dorsal root 12 weeks post-crush from mCherry$^+$ animals (**a**) and hM3Dq$^+$ animals (**b–d**) are shown. Traces of axon growth in the representative images of the mCherry$^+$ and hM3Dq$^+$ animals are shown in **a'** and **b'**, respectively. Higher magnification images of regions in the section from the hM3Dq$^+$ animal are shown in **c** and **d**. ChABC-digestion of CSPG at the DREZ enabled some axons to regenerate across the DREZ 4 weeks after dorsal root crush. There is no difference in the number of mCherry$^+$ axons that regenerated across the DREZ (**e**) or into spinal gray ipsilateral to the crush (**f**) between mCherry$^+$ or hM3Dq$^+$ animals at this time point. As we saw 4 weeks post-crush, at 12 weeks post-crush, some axons extended across the ChABC-treated DREZ in both groups (**a, b, g**), including into white matter (arrows). Virtually no axons extended from control, mCherry$^+$ DRGs into ipsilateral gray matter (**a, a', h**). On the other hand, we saw more axons in ipsilateral dorsal horn from hM3Dq$^+$ DRGs that received daily chemogenetic activation (arrowheads in **c, d, h**). $N = 5$ animals per group at 4 weeks. $N = 6$ animals per group at 12 weeks. Mean ± SEM. Two-way ANOVA and post-hoc multiple comparisons testing using the two-stage step-up method of Benjamini, Krieger, and Yekutieli, $\hat{}p = 0.0144$, $*p = 0.0199$, $**p = 0.0024$, $****p = 0.0001$. Scale bar: 75 μm. Source data are provided as a Source Data file.

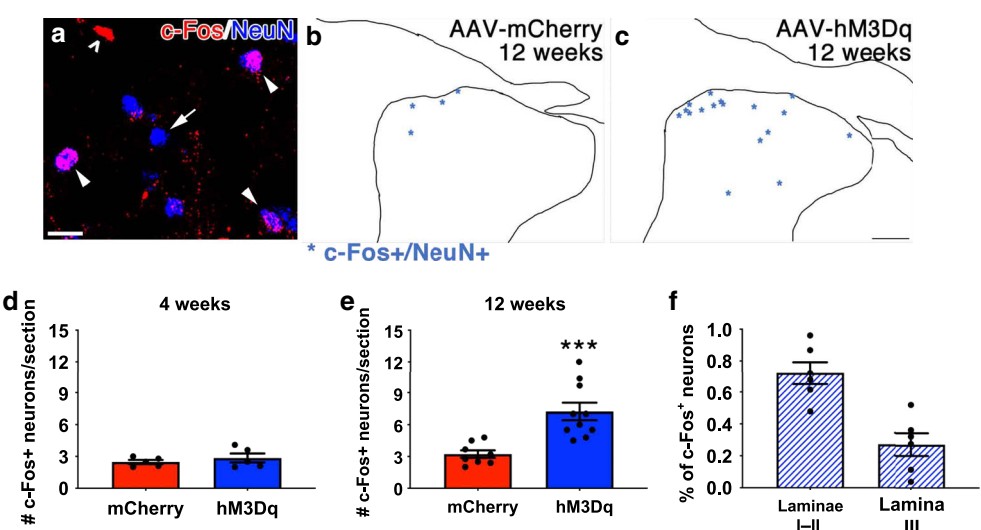

**Fig. 7 Axons that regenerate into spinal gray matter upon chronic, CNO-induced chemogenetic neuron form functional synapses.** Four weeks and 12 weeks after complete C4-T1 dorsal root crush injuries, ipsilateral median and ulnar nerves were isolated and electrically stimulated for 30 min. Animals were sacrificed 1 hr later. Spinal cord sections between C5 and C8 were processed for immunohistochemistry to visualize c-Fos (red) and NeuN (blue), shown in **a**. Not all NeuN$^+$ nuclei were also c-Fos$^+$ (**a**, arrow) and not all c-Fos$^+$ nuclei were in NeuN$^+$ neurons (**a**, open arrowhead). Only c-Fos$^+$/NeuN$^+$ nuclei (**a**, closed arrowheads) were noted and counted. C-Fos$^+$/NeuN$^+$ nuclei in tracings of representative sections from mCherry- or hM3Dq-injected animals 12 weeks after crush are shown as asterisks in panels **b** and **c**. Four weeks after injury, electrical stimulation of the median and ulnar nerves induced c-Fos expression in few neurons within ipsilateral gray matter (**d**). There is no statistical difference between groups. At 12 weeks post-injury, more c-Fos$^+$ neurons were observed in gray matter in animals that received repeated, daily activation of hM3Dq$^+$ DRGs (**e**). The majority of these c-Fos$^+$ neurons were located in more superficial laminae I and II (72.1 ± 0.7%). However, an appreciable percentage (27.3 ± 0.7%) was present in lamina III (**f**). $N = 5$ animals per group at 4 weeks; $N = 8$ mCherry$^+$ animals and 10 hM3Dq$^+$ animals at 12 weeks. Mean ± SEM. Two-tailed unpaired t-test, $***p = 0.0008$. Scale bars: **a** 10 μm; **c** 100 μm. Source data are provided as a Source Data file.

chronic, intermittent neuronal activation of hM3Dq$^+$ DRGs increased their ability to regenerate axons across a ChABC-treated DREZ into the dorsal horn (Fig. 6). Interestingly, neuronal activation-mediated axon regeneration correlated with functional recovery in the grid walking test, a task that is highly dependent upon proprioceptive sensation. We believe that the integration of these regenerated sensory axons into a relevant circuit was responsible for this recovery for several reasons. Large caliber DRG neurons that carry proprioceptive sensory information were the predominant subtype transduced to express hM3Dq (Fig. 2). A much smaller percentage of nociceptive fibers was transduced, corresponding with a lack of a treatment effect in the Hargreaves' thermal sensory test (Fig. 5c). We have evidence that the regenerated axons in the hM3Dq$^+$ animals integrated into spinal circuits, as indicated by the induction of c-Fos in spinal neurons after electrical stimulation of the ipsilateral median and ulnar nerves (Fig. 7). Interestingly, ~27% of those cFos$^+$ neurons were located in lamina III, which contains interneurons that are important for integrating sensory feedback into motor

function[63]. Additionally, we did not observe functional improvements in the hM3Dq$^+$ animals until 6 weeks after crush, a time frame that is inconsistent with sparing of axons following an incomplete crush but is consistent with novel growth and synapse formation mediating the recovery. Moreover, we did not observe regenerated axons in the ipsilateral dorsal horn (Fig. 6f) nor c-Fos$^+$ neurons (Fig. 7d) at the early, 4 week post-crush time point. Unfortunately, the mCherry reporter is fused to hM3Dq and, in our hands, only localized to the soma and portions of axons closer to the soma. Thus, we were not able to reliably assess whether neuronal activation promotes long-distance, dorsal column axon regeneration and/or innervation of the dorsal column nuclei, as has been examined in other studies following other manipulations[64–66].

Chemogenetic technology allows the investigator to easily and remotely activate DRG neurons specifically using a systemic injection of CNO. The duration and timing of stimulation likely affects the efficacy of these treatments. For instance, repeated chemogenetic activation of DRG neurons (5 times per week for

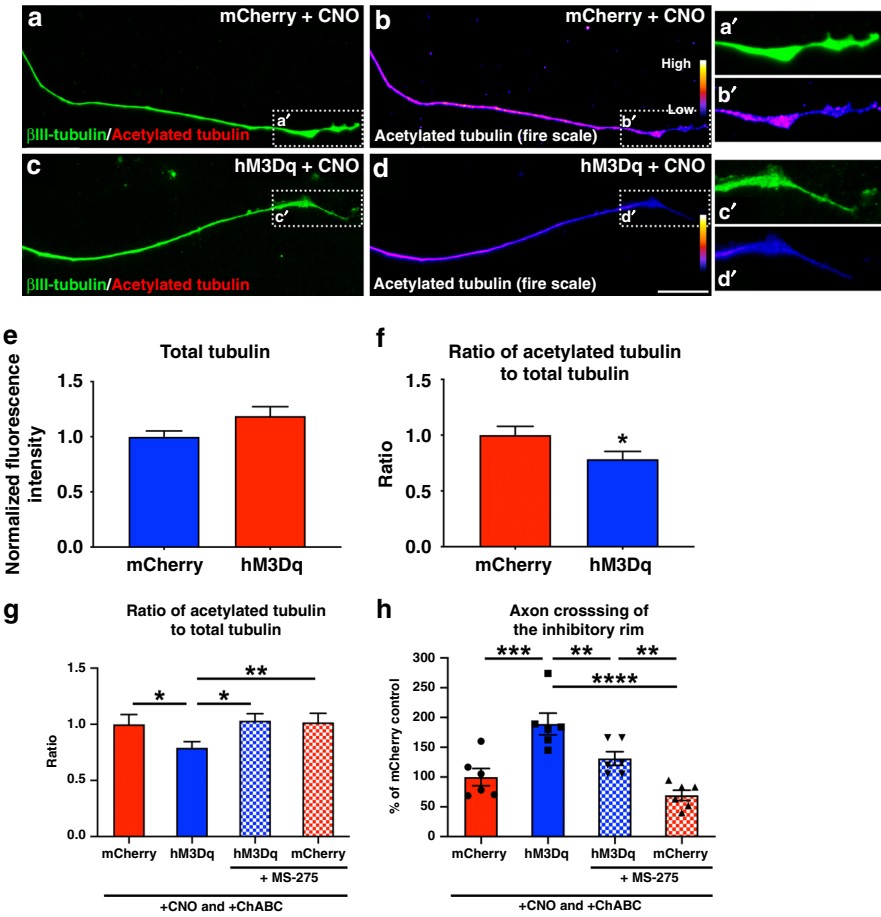

**Fig. 8 Chemogenetic activation of adult DRGs decreases the ratio of acetylated to total tubulin in the distal axon to enhance extension on inhibitory substrates. a–f** Representative images of the distal-most portion of axons from mCherry+ (**a**, **b**) or hM3Dq+ (**c**, **d**) DRGs stained for βIII-tubulin (green) and acetylated tubulin (red in **a** and **c**; standard fire scale pseudo-color in **b** and **d**). Images were taken 24 h after replating and CNO treatment. The distal-most 50 μm portion of axons growing from mCherry+ neurons and chemogenetically activated, hM3Dq+ neurons had a similar fluorescence intensity for βIII-tubulin (**e**). However, the ratio of acetylated tubulin to total tubulin (βIII-tubulin) was significantly smaller in the axons from CNO-activated, hM3Dq+ DRG neurons (**f**), indicating a decrease in stable, acetylated microtubules in axons extending from chemogenetically-activated neurons. **g**, **h** Cultures of mCherry+ or hM3Dq+ DRGs on ChABC-treated aggrecan spots were treated with CNO and either the HDAC inhibitor MS-275 or DMSO control. Chemogenetic activation alone decreased acetylated tubulin levels in the distal axon, compared to mCherry+ controls. Adding MS-275 to activated, hM3Dq+ DRGs increased the ratio of acetylated tubulin to total tubulin in the distal axon end to one that is comparable to mCherry+ (**g**) and abolished the improved ability of activated, hM3Dq+ neurons to cross the inhibitory rim of the aggrecan spot (**h**). N = 72 mCherry+ neurons and 75 hM3Dq+ neurons in **e** and **f**. N = 84 mCherry++vehicle neurons, 88 hM3Dq++vehicle neurons, 74 hM3Dq++MS-275 neurons, and 67 mCherry++MS-275 neurons in **g**. N = 24 aggrecan spots (6 coverslips)/group in **h**. Mean ± SEM. Two-tailed unpaired $t$-test in **f** and one-way ANOVA and post-hoc multiple comparisons testing using the two-stage step-up method of Benjamini, Krieger, and Yekutieli in **g** and **h**. *$p < 0.05$ ($p = 0.0434$ in **f**; mCherry+vehicle vs. hM3Dq+vehicle $p = 0.0436$, hM3Dq+vehicle vs. hM3Dq+MS-275 $p = 0.0250$ in **g**), **$p < 0.01$ (hM3Dq+vehicle vs. mCherry+MS-275 $p = 0.0035$ in **g**; hM3Dq+vehicle vs. hM3Dq+MS-275 $p = 0.0071$, hM3Dq+MS-275 vs. mCherry+MS-275 $p = 0.0045$ in **h**), ***$p = 0.0002$, ****$p < 0.0001$. Scale bar: 10 μm. Source data are provided as a Source Data file.

2 weeks) following sciatic nerve injury resulted in greater axon regeneration than a single activation[67]. However, continuous electrical stimulation for 2 weeks does not[68]. Whether the difference is due to the stimulation pattern itself (repeated, intermittent stimulation vs. continuous stimulation) or whether electrical stimulation activates pathways that are different than chemogenetic activation remains to be investigated fully. Moreover, the optimal pattern of stimulation required to promote growth is still not well understood. Optogenetics or electrical stimulation would be a better means to address this important question, given that one is better able to regulate frequency and duration of activation with these techniques. Nevertheless, our studies support the notion that recurrent, intermittent neuronal activation over a prolonged period of time can enhance axon

regeneration; we saw more functional axonal regeneration after 12 weeks of daily, neuronal activation than 4 weeks.

How does DRG activation enhance sensory axon regeneration? Interestingly, we found that even though neuronal activation does not impact total tubulin levels, it does shift the distal end of extending axons from being rich in acetylated tubulin, a marker of stable microtubules, to being abundant with tyrosinated tubulin, a hallmark of labile microtubules (Figs. 8 and 9). This phenotype is consistent with neuronal activation enriching the leading end with a dynamic, labile microtubule array, recapitulating what occurs in juvenile axons. It is known that pathfinding depends on labile microtubules, which allow the growth cone to steer the extending axon appropriately in response to attractive or repulsive guidance cues[69–72]. Additionally, labile microtubules

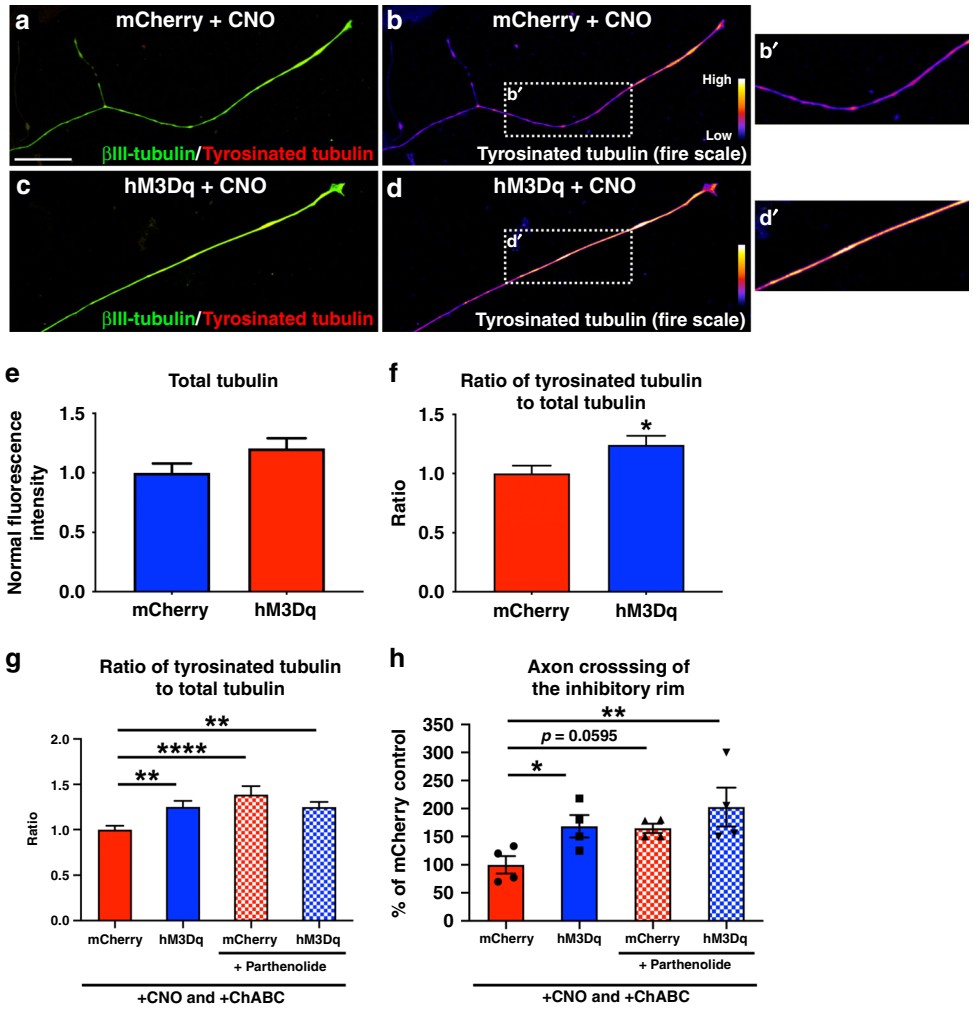

**Fig. 9 Chemogenetic activation of DRGs increases the ratio of tyrosinated to total tubulin in the distal axon to improve growth on inhibitory substrates. a–f** Representative images of the distal-most portion of axons from mCherry+ (**a**, **b**) or hM3Dq+ (**c**, **d**) DRGs stained for βIII-tubulin (green) and tyrosinated tubulin (red in **a**, **c**; standard fire scale pseudo-color in **b**, **d**) 24 h after replating and CNO treatment. There was no difference in the fluorescence intensity for βIII-tubulin in the distal-most 50 μm of axons growing from mCherry+ neurons and chemogenetically activated, hM3Dq+ neurons (**e**). However, the ratio of tyrosinated tubulin to total βIII-tubulin was higher in the axons from CNO-activated, hM3Dq+ DRG neurons (**f**), indicating an increase of labile microtubules in axons extending from chemogenetically-activated neurons. **g**, **h** Cultures of mCherry+ or hM3Dq+ DRGs on ChABC-treated aggrecan spots were treated with CNO and either parthenolide to inhibit tubulin carboxypeptidase or DMSO control. Adding parthenolide to mCherry+ DRGs increased the ratio of tryosinated tubulin to total tubulin in the distal axon end (**g**) and improved their ability to extend axons across the inhibitory rim of the aggrecan spot. The number of axon crossings from parthenolide-treated, mCherry+ DRGs was similar to activated, hM3Dq+ neurons (**h**). $N = 86$ mCherry+ neurons and 87 hM3Dq+ neurons in **e**, **f**. $N = 87$ mCherry++vehicle neurons, 72 hM3Dq++vehicle neurons, 72 mCherry++parthenolide neurons, and 73 hM3Dq++parthenolide neurons in **g**. $N = 16$ aggrecan spots (4 coverslips)/group in **h**. Mean ± SEM. Two-tailed unpaired t-test in **f** and one-way ANOVA and post-hoc multiple comparisons testing using the two-stage step-up method of Benjamini, Krieger, and Yekutieli in **g**, **h**. *$p < 0.05$ ($p = 0.0195$ in **f**; mCherry+vehicle vs. hM3Dq+vehicle $p = 0.0480$ in **h**), **$p < 0.01$ (mCherry+vehicle vs. hM3Dq+vehicle $p = 0.0066$, mCherry+Vehicle vs. hM3Dq+Parthenolide $p = 0.0071$ in **g**; mCherry+Vehicle vs. hM3Dq+parthenolide $p = 0.0062$ in **h**), ****$p < 0.0001$. Scale bar: 10 μm. Source data are provided as a Source Data file.

also play a critical role in the microtubule engorgement of the growth cone, a crucial step for consolidating a growth cone into a new axon[73,74]. Therefore, neuronal activation may enable the extending axon to navigate through environments by increasing dynamic microtubules in the growth cone[70,74].

Interestingly, we found that experimentally increasing the levels of tyrosinated tubulin in the axon with parthenolide recapitulates the positive effect of neuronal activation on growth on an inhibitory substrate in vitro (Fig. 9g, h), and that experimentally increasing the levels of acetylated tubulin with MS-275 diminished the neuronal activation-mediated growth on the same inhibitory substrate (Fig. 8g, h). These results are consistent with studies indicating that suppression of tubulin detyrosination was

shown to be beneficial to peripheral nerve regeneration[51]. At first glance, these results seem to suggest that it is not the dynamics of the microtubules that are relevant so much as it is the absence of the post-translational modifications that normally accompany stability. However, in living cells, various microtubule-related proteins interact differently with microtubules on the basis of their content of these different tubulins and this may account for the positive benefits to axon regeneration. For example, we have found that knocking down expression of fidgetin, a protein that pares the labile domain of microtubules, results in increased axon regeneration after dorsal root crush[24], and does so by elevating the levels of microtubules that are less acetylated and more tyrosinated[75,76]. Thus, knocking down fidgetin may augment

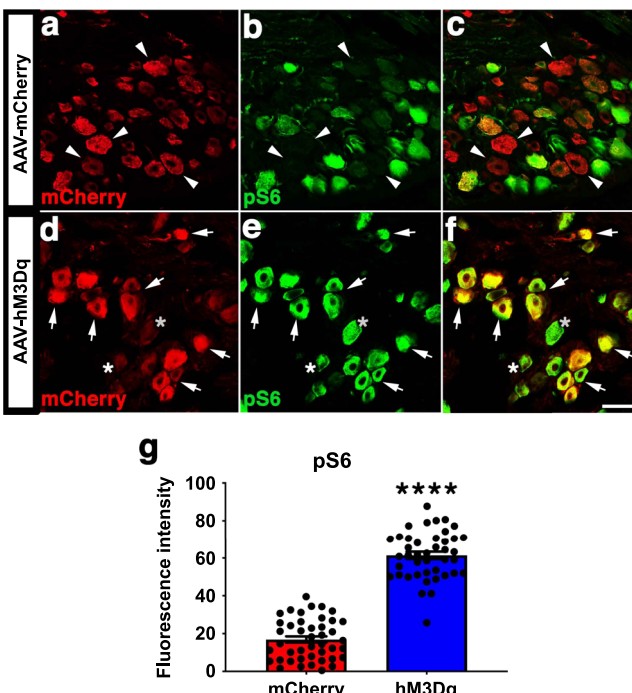

**g** pS6

**Fig. 10 Chemogenetic neuronal activation increases mTOR activity.** DRGs injected with either hM3Dq or mCherry. All animals received CNO. One day after CNO administration, sections were stained for p-S6, a marker of mTOR activation. While some mCherry+ neurons had detectable p-S6, many did not (**a–c**, arrowheads). However, the vast majority of hM3Dq+ DRGs had high levels of p-S6 (**d–f**, arrows), even neurons that weakly expressed hM3Dq (asterisks). (**g**) Quantification of p-S6 fluorescence intensity. $N = 43$ mCherry+ neurons and 42 hM3Dq+ neurons. Mean ± SEM. Two-tailed unpaired t-test, ****$p < 0.0001$. Scale bar: 70 μm. Source data are provided as a Source Data file.

axon regeneration by mimicking the effects observed here with neuronal activation, or neuronal activation might act through a pathway that suppresses fidgetin or a related protein. These concepts will be explored in future experiments.

While neuronal activation did not impact the total number of axons that extended across the DREZ into white matter, it did increase the number of axons that turned off of these white matter tracts to extended in gray matter (Fig. 6b–d, h). This required that the axon modify the orientation of growth and turn to extend from white matter to innervate the dorsal horn. While white matter can support axon regrowth that runs parallel to the geometry of the fiber tract, presumably along aligned astrocytic processes[11,77,78], nonparallel regrowth is largely inhibited[79,80]. Because axon turning is dependent upon having tyrosinated tubulin within the growth cone[70,72], the increase in tyrosinated tubulin in axons of activated, hM3Dq+ neurons may enable their ability to turn and extend in a non-parallel fashion towards the dorsal horn.

Our findings add to a growing body of data that challenge the view that microtubule stabilization is crucial for axonal regeneration in the CNS[81–84], and instead support the view that successful axonal regeneration involves shifting a predominantly stable microtubule array into a more dynamic one[24]. Indeed, peripheral axons have been shown to regenerate better when their microtubules are less acetylated and more tyrosinated[51,85,86], and the same is likely to be true of centrally-projecting axons. We suspect that pharmacologic agents that stabilize microtubules may have provided limited success in augmenting axon regeneration in previous studies for reasons that did not tap into

normal physiological mechanisms. For example, stabilization of microtubules in neurons prevents axonal retraction and hence the drugs may enable the axon's tip to push through normally inhibitory environments. However, these effects do not reflect how axonal growth normally occurs, given that reduced microtubule dynamics contributes to the developmental decline of axon growth capacity[49,87], and these drugs may introduce troubling side-effects in humans (e.g., neuropathy[88]). It may also be that these drugs stabilize microtubules in non-neuronal cells to diminish glial scar formation[89] to indirectly improve axon regeneration.

The alterations in microtubules indicated by our studies may be a downstream component of pathways that promote axon regeneration, and neuronal activation may be upstream of those pathways. Our studies indicate that activated DRG neurons have more mTOR activation than controls (Fig. 10). mTOR is known to protect microtubules from microtubule stabilization[90] and to regulate the interaction of microtubules with CLIP170, a protein that associates with the tips of dynamic microtubules[91], so it may be via these effects that neuronal activation alters microtubules. Moreover, previous studies showed that stimulation promotes corticospinal tract axonal sprouting and synapse formation via upregulation of mTOR and Jak/Stat signaling[92] and that inhibiting tubulin detyrosination to keep microtubules dynamic mimicked the axon growth-promoting effects of sustained GSK3 activity[51,93]. Collectively, these results support the view that various different pathways beneficial to nerve regeneration may converge on a common downstream effect, which is to shift the microtubule array of the axon, especially the distal axon, toward being less post-translationally modified and more dynamic.

It is also relevant that stimulation has been shown to increase neurotrophin expression in both neurons and Schwann cells[94–98], which plays an important role in activity-induced regeneration[94,99]. Therefore, it is likely that multiple mechanisms are working synchronously to enhance regeneration after neuronal activation.

Axon that regenerate do not always form functional synapses[42,43,100]. Thus, the positive effects of neuronal activation in this study are likely not limited to enhancement of axon extension. Many studies of developing circuits have demonstrated that activity drives synapse formation and plays a critical part in the maintenance of synaptic connections[101–104]. On the other hand, interfering with neuronal activity has a negative effect on synapse number. Genetic deletion of proteins required for neurotransmitter release dramatically reduces the number of synapses in the cortex and neuromuscular synapses[105,106]. Interestingly, neuronal activation can affect not only the neuron that is being activated but also the putative post-synaptic neuronal partner. Studies have shown that growth cones of growing axons undergo constitutive vesicle cycling and release neurotransmitter, even prior to synapse formation[107,108]. This presynaptic release of neurotransmitter, which can be achieved by neuron activation, can induce dendritic protrusion[109,110], spine formation[111,112], and synapse formation[113,114]. This activity-mediated dendritic growth is dependent upon the activation of postsynaptic receptors[112,115]. Thus, it is possible that chemogenetic activation of hM3Dq+ DRG neurons not only enhanced their regenerative capacity but also indirectly affected neurons within the spinal cord, making those neurons more receptive to being synapsed upon by the regenerated axons. How neuronal activation affects synapse formation after injury will be an interesting avenue to explore in future studies.

Another consideration is that ChABC alone promotes plasticity. Administration of ChABC allows for digestion of CSPGs within the perineuronal nets, structures that surround mature neurons in the CNS, including in the spinal cord, and limit

plasticity[116–122]. Additionally, ChABC has been shown to promote collateralization of uninjured fibers that may have functional implications, depending on the context[6,42,123–126]. It is likely that ChABC opens up a window of plasticity that makes the spinal cord more amenable to the effects of chemogenetic activation-mediated axon regeneration. Moreover, we injected ChABC at a single time point. While a single injection of ChABC remains bioactive for over a week[127], it is clear that plasticity can continue with long-term delivery of ChABC via viral vectors[128–130]. It will be interesting to determine the optimal time frame of ChABC administration (e.g., with a regulatable gene therapy to deliver ChABC[7]) together with chemogenetic activation. It is possible that more prolonged enzyme delivery would further enhance activation-mediated growth and/or the functional integration of this growth.

In conclusion, chemogenetically activating hM3Dq+ DRGs after dorsal root crushes enhances functional axonal regeneration across the DREZ, and does so in a manner that is dependent upon changes in the post-translational modifications of microtubules in the distal region of the axon. In future studies, it will be important to focus on optimizing the conditions for more efficient regeneration as well as explore other mechanisms that underlie activity-enhanced axonal regeneration. For now, the present studies add to a growing body of evidence that microtubules—specifically characteristics of microtubules associated with dynamics and not stability—are a common downstream target of various pathways that promote regeneration.

## Methods

**Animals.** Adult female Wistar Rats weighing between 225 to 250 g were purchased from Charles River. Animals were maintained in 12 h light/dark cycle, given unlimited access to food and water. All animal use was conducted in accordance with National Institutes of Health guidelines for animal care and use and approved by the Drexel University Institutional Animal Care and Use committee.

**In vitro analysis of DRG neurite growth.** DRGs from adult Wistar rats were dissected and dissociated as described previously[15,34]. Briefly, DRGs were dissected from the spinal cord on ice and were kept in HBSS on ice. After trimming roots, the DRGs were treated with collagenase (200 Units/ml, Worthington Biochemical Corporation) and neutral protease (25 Units/ml, Worthington Biochemical Corporation) at 37 °C for 30 min. Enzymatically digested DRGs were washed several times with HBSS and then gently triturated in culture media (containing Neurobasal A, B-27, GlutaMax and Gibco's Antibiotic-Antimycotic) into a single-cell suspension. The dispersed cell suspension was transferred to a new 1.5 ml tube. After two rounds of low-speed spins to remove satellite cells, the DRG neurons pellet was resuspended into culture media at a density of 8000 neurons per milliliter. The DRG neurons were plated on a poly-D-Lysine (0.1 mg/ml, Sigma-Aldrich) coated 6 well plates and incubated with AAV5-hSyn-mCherry or AAV5-hSyn-hM3Dq-mCherry (Duke University's viral vector core; 3 × 10^9 GC/ml). The dissociated DRG neurons were incubated for 72 h (to allow for transduction and transgene expression) before detaching them from the 6-well plate by gently scrapping them off from well bottom with a cell scrapper.

To examine how neuronal activation affects neurite growth on inhibitory substrates in vitro, the transduced, dissociated neurons were seeded on aggrecan-laminin spot gradient coverslips in 24-well plates at a density of 4000 cells per coverslip. The coverslips with aggrecan spots were prepared one day before replating cells. Glass coverslips were coated with poly-D-Lysine (0.1 mg/ml) and nitrocellulose. Then four drops of 2 μl aggrecan (0.4 mg/ml, Sigma-Aldrich) were spotted on each coverslip. The coverslips were allowed to air dry in the hood and then incubated with 10 μg/ml of laminin (Sigma-Aldrich) and 0.5 unit/ml of ChABC (Sigma-Aldrich) on the day of replating for 6–8 h. 10 μM of CNO was applied to the cell culture media. Five days after replating, the cell cultures were fixed with 4% PFA in 0.1 M PBS and then processed for immunocytochemistry using an anti-βIII tubulin antibody (Sigma-Aldrich, 1:1000) to label neuronal microtubules. The next day, the coverslips were rinsed in PBS and then incubated in AlexaFluor488-conjugated secondary antibody for 2 h at room temperature. The coverslips were rinsed in PBS for 3 times and mounted onto glass slides using FluorSave (EMD Biosciences), and number of neurites crossing the aggrecan rim were counted per spot using an Olympus BX51 fluorescence microscope. The average number of crossing neurites per spot was calculated and normalized to the value of the AAV-hSyn-mCherry + ChABC group. At least three independent experiments were conducted.

**Surgical procedures.** Animals were allowed to acclimate for at least 1 week after arrival before any surgical procedures. During surgeries, animals were anesthetized with isoflurane and kept on a heating pad to prevent hypothermia and all surgical procedures were performed under aseptic technique. After all surgical procedures, a piece of synthetic matrix membrane (BioBrane; UDL Laboratories) was placed over the exposed cord and DRGs to protect the spinal cord and to minimize scar accumulation on the dura. The overlying musculature was closed using 5-0 sutures, and the skin was stapled shut with surgical staples (Fine Science Tools). Animals were given subcutaneous injections of 5 ml lactated Ringer's solution to prevent dehydration, ampicillin (200 mg/kg) as the antibiotic, and slow-release buprenorphine (0.1 mg/kg; ZooPharm) for pain. Animals were placed on a thermal barrier until fully recovered from anesthesia.

**Intraganglionic AAV injections.** AAV5-hSyn-hM3Dq-mCherry and AAV5-hSyn-mCherry were obtained from Duke University's viral vector core. To inject AAV5-hsyn-hM3Dq-mCherry or the control AAV5-hSyn-mCherrry into the right C6, C7, and C8 DRGs, a hemilaminectomy was performed at the corresponding vertebral segments. Enough of the bone was removed to fully expose the C6-C8 DRGs and associated dorsal roots. Viruses were mixed with sterile PBS to reach the same final titer of 3.6 ×10^9 GC/μL. After dilution, viruses were loaded into a beveled-tip glass micropipette that was mounted on the needle of a 10 μl Hamilton syringe (Product #07939-10). The glass micropipette was carefully inserted into the exposed ipsilateral C6-C8 DRGs and 1 μl of either AAV5-hsyn-hM3Dq-mCherry or control AAV5-hSyn-mCherry was injected into each DRG at a rate of 0.1 μl per minute. The glass micropipette was left in place for 1–2 min before removal to prevent reflux.

**Dorsal root crush injury and ChABC injection.** Two to three weeks after AAV5 injections, rats were anesthetized with isoflurane and the cervical spine was carefully re-exposed again. An additional laminectomy was performed at C5 and T1 to expose the C5 and T1 dorsal roots. As we did in[15,24], the right dorsal roots from C5 to T1 were crushed with fine #5 forceps (Fine Science Tools). To do this, a small slit was made in the dura immediately caudal to each dorsal root, and one tine of the forceps was carefully placed under the dorsal root, halfway between the distal end of the dorsal root and the DRG. The forceps were squeezed for 10 s to crush the root. Particular care was taken to not pull the root or damage the surrounding tissue. Each dorsal root was crushed three times to ensure the injury completeness of the afferent axons. In our experience, this method virtually always results in a complete dorsal root crush injury; we rarely observe animals with incomplete injuries where axons are found across the DREZ in the absence of any manipulation. Immediately after dorsal root crushes were completed, 1ul of ChABC (50 U/ml; Sigma-Aldrich) was injected into the C6-C8 DREZ of all animals (3 injections along the C6-C8 rostral to caudal axis) using a beveled glass micropipette. To confirm completeness of the crush, acutely after the injury, we looked for a characteristic clasping of the front right paw, a phenotype that is not observed in animals with incomplete dorsal root crush injuries.

**CNO administration.** Animals were injected with 1 mg/kg CNO (s.c., 1 mg/mL in 0.5% DMSO/0.9% saline) every evening starting the day after the dorsal root crush to the day before sacrificing for the duration of the experiment (i.e., up to 12 weeks post-crush). All behavioral analyses were performed in the morning, prior to that day's CNO administration.

**Behavioral analyses.** Animals injected with mCherry or hM3Dq (n = 7 animals per group) were assessed using the following behavioral tests:

Gridwalk test: The gridwalk test examines how well animals correctly place their forepaw on a rung of a grid, which requires both motor and proprioceptive function. Briefly, the paw placement efficiency was assessed as the animals walked on an elevated, plastic coated wire mesh grid with 3 cm × 3 cm openings. We placed a mirror to be able to observe paw placement on the grid from any angle during video recording. At each trial, one animal was placed on the grid for 2 min and allowed to walk freely across the space. Each forelimb was scored for the total number of steps and the total number of missteps. A misstep was counted when the limb fell through the grid. The total number of steps and missteps were added together to obtain the total number of placements. The percent efficiency was calculated by dividing the number of correct steps by the total number of placements. Animals were trained to walk on the wire mesh grid before baseline testing. The baseline paw placement efficiency was obtained before animals were subjected to any surgery. After dorsal root crushes, the percent of correct paw placements were assessed weekly until week 12.

Von Frey Filament Test: The Dynamic Plantar Aesthesiometer (Ugo-Basil) was used to perform the Von Frey filament assessment of mechanical hyperalgesia on the plantar surface of rats. All animals were habituated to the testing apparatus at least once prior to obtaining a baseline score. Behavioral testing was conducted before any surgeries to establish the baseline responses and then weekly after dorsal root crush injuries for 12 weeks. On each testing day, animals were placed in a clear Plexiglass chamber with paws placed on a perforated metal platform. A movable force actuator was positioned below the metal platform. After an acclimation period of time, increasing force was applied to the center of the plantar surface of

the tested paw by a Von Frey–type 0.5 mm filament with a maximum force of 50 g. The Dynamic Plantar Aesthesiometer (DPA) automatically detected and recorded latency time, and actual force at the time of paw withdrawal reflex. Vocalization, licking, grading as well as withdrawal were considered a positive response. Animals that did not withdraw the paw at the maximum force or only withdrew the paw after the paw was lifted up by the filament was considered a negative response and the response threshold was recorded as the maximum 50 g. On every test day, 5 trials were performed on each paw for each animal and paw testing order was determined randomly to minimize an order effect. The withdrawal threshold was averaged among 5 trials for each animal at each testing time point.

Hargraeves' test: Animals were habituated to the Ugo-Basile Plantar Heat test apparatus at least once prior to baseline testing, and baseline testing was performed before any surgical procedures. The changes in the thermal sensation after injuries were measured weekly for 12 weeks after dorsal root crush injuries. On each testing day, animals were allowed to acclimate for at least 10 mins in their testing chamber. Afterwards, an infrared heat source was placed directly beneath the center of the plantar surface of the paw to be tested. The noxious infrared light beam was applied to the plantar surface until animals withdrew their paws and the paw withdrawal latency was recorded. The infrared heat source automatically shut off at 30 s to avoid tissue damage. In addition to paw withdrawal, observation of any responses to the thermal stimulus, including vocalizations, licking the paw, turning the head to look at the stimulus were all considered positive responses and the response latency were recorded. On each testing day, 5 trails were performed for each paw, with at least 1 min intervals between each trial, and the paw testing order was randomly determined. The withdrawal latency for each animal at each time point was calculated by averaging the withdrawal latencies from 5 trials.

**Electrical stimulation to induce c-Fos expression.** At 4 weeks ($n = 5$ animals per group) and 12 ($n = 8$–10 animals per group) weeks post dorsal root injuries, animals injected with AAV5-hSyn-mCherry or AAV-hSyn-hM3Dq-mCherry in DRGs were anesthetized with ketamine and xylazine. The median and ulnar nerves in the right forelimb were carefully dissected free and placed on a bipolar hook electrode for stimulation. The nerve was stimulated for 30 min using 300 ms stimulation pulse-trains of 1 mA amplitude, 0.1 ms pulse duration, and 100 Hz frequency. Mineral oil was applied several times during the stimulating period to ensure the nerves did not dry out. One hour after stimulation, animals were perfused with 4% PFA and prepared for immunofluorescent staining. One of every 6 spinal cord sections from C6 to C8 were stained with antibodies against c-Fos and NeuN. The total number of c-Fos$^+$ nuclei in each spinal cord section and the number of c-Fos$^+$/NeuN$^+$ double-stained nuclei in spinal cord dorsal horn were counted.

A summary of the in vivo paradigm is shown in Fig. 4.

**Histology.** Four weeks ($n = 5$ per group) or 12 weeks ($n = 8$–10 animals per group) after dorsal root crushes, animals were euthanized by injection of Euthasol and perfused transcardially with ice cold 0.9% saline, followed by 4% paraformaldehyde in 0.1 M phosphate buffer (pH 7.4). The spinal cord from C6 to C8, with dorsal roots attached, and DRGs were removed, post-fixed in 4% PFA at 4 °C overnight and cryoprotected in 30% sucrose in 0.1 M PBS at 4 °C for at least 48 h. Tissue blocks were embedded in OCT and quick frozen on dry-ice. The spinal cord with attached dorsal roots were transversely sectioned at 25 μm using a cryostat and collected in PBS. DRGs were cryosectioned at a thickness of 20 μm and mounted directly onto gelatin-coated slides. For staining, sections were blocked in 5% normal goat serum and 10% BSA with 0.1% Triton X-100 in PBS for 1 h and then incubated with primary antibodies at 4 °C overnight. The primary antibodies used were anti-mCherry (1:500; Abcam), anti-p75 (1:1000; Millipore), anti-c-Fos (1:1000; Santa Cruz), anti-NeuN (1:100; Millipore), anti-NF-200 (1:500; Sigma), anti-IB4 (1:1000; Sigma), anti-calcitonin gene related peptide (CGRP; 1:1000; Peninsula Laboratories International), and anti-p-S6 (1:800; Cell Signaling). The next day, sections were rinsed three times, incubated with appropriate AlexaFluor-conjugated secondary antibodies [goat anti-mouse Alexa 488 (Invitrogen, A11001); goat anti-mouse Alexa 594 (Invitrogen, A11005); goat anti-rabbit Alexa 488 (Invitrogen, A11008); goat anti-rabbit Alexa 594 (Invitrogen, A11012), all at 1:500] for 2 h, and washed in PBS. After staining, sections were coverslip-mounted with FluorSave (EMD Chemical) and photographed using a Leica DM5500B epifluorescence microscope and a Leica SP8 confocal microscope using Slidebook 6 and LAS X, respectively.

**Quantification of c-Fos induction in DRGs administered CNO.** Four weeks after intraganglionic injections of AAV-hM3Dq-mCherry or -mCherry control ($n = 5$ animals per group), animals were injected with CNO and perfused 2 h later. DRG sections were stained with antibodies against mCherry and cFos, an established means to confirm chemogenetic neuronal activation[39,40,131]. While blinded to treatment group, the percentage of mCherry$^+$ neurons that were also c-Fos$^+$ was calculated (6 sections analyzed per animal).

**Quantification of axon growth across the DREZ.** Similar to what we did in[15], coronal spinal cord sections ($n = 6$ animals per group) with the dorsal roots attached were immunostained with antibodies against mCherry to visualize

regenerated axons from AAV5-hSyn-mCherry or -hM3Dq transduced DRG neurons and p75 to visualize Schwann cells within the dorsal root. The DREZ was identified as the interface between the p75$^+$ dorsal root and the p75$^-$ spinal cord. Z-stack confocal microscope images presented with maximum intensity projection were used for quantification. While blinded to treatment group, mCherry$^+$ axons that extended distal to the p75$^-$-defined DREZ and into gray matter were counted using ImageJ 1.47 V and were binned into 50 μm intervals.

**Microtubule analysis.** To prepare DRG samples for immunofluorescence visualization and quantification of microtubules, dissociated DRG neurons were transduced to express mCherry or hM3Dq, as described above, and replated 3 days later on poly-lysine for 24 h. Prior to preparation for immunofluorescence, for the purpose of examining the functional relevance of microtubule acetylation and tyrosination, some cultures were treated with the histone deacetylase inhibitor MS-275 (1 μM), the tubulin carboxypeptidase inhibitor parthenolide (5 nM), or DMSO vehicle. Cultures were pre-extracted for 4 min in PHEM (PIPES, HEPES, EDTA, MgCl2) buffer containing 0.1% Triton X-100. PHEM buffer preserves microtubules and Triton X-100 releases free tubulin[132]. After pre-extraction, DRG cultures were then fixed in a solution containing 4% paraformaldehyde and 0.2% glutaraldehyde[133]. After blocking with normal goat serum and BSA, the fixed cultures were incubated with antibodies against βIII-tubulin (to indicate the total neuronal microtubule mass) and acetylated tubulin (Sigma) or tyrosinated tubulin (Sigma) and then appropriate secondary antibodies. All staining was performed at the same time. Imaging was conducted using a Leica DM5500B epifluorescence microscope and a Leica SP8 confocal microscope using Slidebook 6 and LAS X, respectively. Exposure times for each antibody were kept consistent across groups. Approximately 100 axons were imaged per dish for each treatment condition (repeated in 3 independent experiments). Using ImageJ, the region 50 μm from the distal axon tip (growth cone) was traced around axons and the integrated signal intensity for βIII-tubulin and acetylated or tyrosinated tubulin in that region was recorded. The ratio of fluorescence intensity for acetylated to total tubulin or tyrosinated to total tubulin was then calculated[134].

**Statistics and reproducibility.** All animals and neuronal cultures used in these studies were randomly assigned to groups prior to treatment or any experimental manipulation. Sample sizes (also noted in the relevant figure legends) were determined based upon our previously published studies using similar techniques[14,15,133]. Culture experiments were conducted in triplicate and animal experiments were performed in two independent cohorts, with controls in each. All analyses were performed while blinded to treatment group. For pairwise comparisons, a two-tailed student's t-test was performed. A two-way ANOVA and post-hoc multiple comparisons testing using the two-stage step-up method of Benjamini, Krieger, and Yekutieli[135] were performed to ascertain differences between groups over time. A one-way ANOVA and post-hoc multiple comparisons testing using the two-stage step-up method of Benjamini, Krieger, and Yekutieli were performed to assess for differences between three or more groups in which time was not a consideration. A p-value < 0.05 was considered significant. All statistical tests were performed using GraphPad Prism 8.

**Reporting summary.** Further information on research design is available in the Nature Research Reporting Summary linked to this article.

## Data availability

All data supporting the findings of this study are available from the corresponding author upon reasonable request. The source data for Figs. 1d, 2j, 3g, 5a–c, 6e–h, 7d–f, 8e–h, 9e–h, 10g, Supplemental Fig. 1a, b are provided in the separate Source Data file. Source data are provided with this paper.

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

## Acknowledgements

Funding for these studies was provided by NIH R01 NS085426, NIH R01 NS106908, and NIH R01 NS111761 to V.J.T.; R01 NS28785 and R01 NS118117 to P.W.B.; the U.S. Army Medical Research and Materiel Command (W81XWH1210379) to P.W.B.; the Craig H. Neilsen Foundation (259350) to P.W.B. We thank Andrew Matamoros for providing technical guidance and the Marion Murray Drexel University Spinal Cord Research Center for use of its core facilities.

## Author contributions

D.W. and V.J.T. designed the experiments. D.W., Y.J., T.M.S, and A.H. performed the research. P.W.B. provided technical and conceptual guidance. D.W., Y.J., P.W.B., and V.J.T. analyzed the data and wrote the paper.

## Competing interests

The authors declare no competing interests.
