## [Peer Review File · Nature Communications]

Reviewers' Comments:

Reviewer #1:

Remarks to the Author:

The paper by Tom and co-authors examines sensory axon regeneration into the spinal cord of adult rats using a novel combination approach aimed at stimulating new growth and functional connectivity in the spinal dorsal horn. They used a combination of repeated chemogenetic activation of adult sensory neurons in the dorsal root ganglia, DRG, (by in vivo AAV transduction of the excitatory DREADD receptor hM3Dq and activation with daily CNO injections) and enzymatic modulation of the inhibitory environment at the dorsal root entry zone, DREZ (by intraspinal injection of the neuroplasticity-promoting agent chondroitinase ABC, ChABC). They present evidence of anatomical regrowth, functional connectivity and recovery of in a grid walk task with long term (12 weeks) but not short term (4 weeks) treatment. These are novel and important findings. The grid walk data is particularly convincing (although may not be related only to dorsal horn reinnervation). Some important control groups were not included, so in the absence of these a more in-depth analysis of axonal growth would be appreciated to more fully convince the observed effects are due to the specific treatments. Also, important methodological detail is missing and needs clarification.

1. What happens to the collaterals of the large diameter projections that ascend in the dorsal columns and are important for proprioception? As this population is labelled with mCherry the authors should look at this projection in more rostral segments and, importantly, in their terminal fields in the brainstem dorsal column nuclei. Previous studies using a similar injury (C5-C8 dorsal root crush), treatment paradigm (AAV DRG delivery) and long term (12 week) time course showed regrowth up to the C1 spinal cord and medulla (e.g. Cheah et al 2016 J Neurosci 36, 7286; this paper should be cited); others have shown long-distance ascending sensory axon growth to the medulla but no innervation of the nuclei (e.g. Tuszynski work). It seems important that this projection, not just dorsal horn innervation, is examined.

2. Intraganglionic AAV injections involve direct injections into the C6-C8 DRG (one microliter injected with a glass micropipette, then left in place for 1-2 min before removal). This is performed 2 weeks prior to the dorsal root crush injury, so in effect is likely to cause a "conditioning injury". A control group should have been included which has no AAV injection, to see how much the injection procedure itself affects DRG growth, and then make the comparison +/- DREADD, to see how much this is amplified by DREADD activation. In absence of this, the authors could examine the DRG (immunohistochemically or biochemically) for injury and growth-associated markers and/or expression of activated pathways involved in neuronal activation-mediated growth e.g. mTOR, JAK/STAT, tubulin-modifying proteins, ATF3, GAP-43). Some analysis of the DRG cell body response across the different treatment groups would significantly strengthen the manuscript.

3. Methodology detail:

- Study design, blinding and randomisation etc not specified.
- Frustratingly no n numbers are provided – how many animals were perfused at 4 weeks and 12 weeks post-injury? How many animals/group were used for terminal electrical stimulation studies (to induce cFos) at the 4 week and 12 week time points? In the experimental design section (P14) only minimal information is provided e.g. "Sample sizes were determined based upon our previous published studies using similar techniques", but then no sample size numbers are actually provided.
- CNO administration – add more detail, how many consecutive days did animals receive CNO, what was the total amount injected? State early in the manuscript (e.g the abstract) the time frame and treatment regime e.g. daily CNO injections over 12 weeks post-rhizotomy?
- No mention of any adverse effects of daily CNO injections over 12 weeks. This must be included.
- More control groups would have strengthened the data i.e. a no-ChABC group; a no-CNO group. Justify why these groups were not included and add some discussion on interpretation of the results in absence of these groups.

- A timeline figure should be included, to clearly show the timings of AAV administration, CNO dosing, rhizotomy, ChABC injection, electrical stimulation and behavioural testing time points.
- P13 "mCherry+ axons...were counted using ImageJ" – again, no detail provided - how many sections/animal, what regions were analysed i.e. only the DREZ transition zone, superficial dorsal horn, deep dorsal horn (which lamina?), how many ROI and how were ROI defined?
- How was the "completeness" of injury confirmed? There is always some doubt with a crush injury that there will be some spared axons. Claspings of the paw does not confirm completeness, this can only be confirmed anatomically e.g. by tracing and absence of terminal labelling in the cuneate nuclei.

4. Figure 3: Provide a more detailed/comprehensive picture of the extent of regeneration. Multiple dorsal roots were crushed (from C5 to T1), yet only two images are shown and it is not clear which segmental level they are from, or whether the pattern/extent of axon regeneration differs at different levels? Are the counts reflective of the overall number across the entire extent of C5-T1? A rostro-caudal reconstruction across C5-T1 would be informative. Also, more detail on which fibre type is growing across the DREZ would be informative – they are presumably mostly the large-diameter NF200-positive axons given that a higher percentage of these were transduced than C fibres, however given that ~30% transduced neurons were nociceptive neurons and that cFos expression in fig 4 looks to be more localised in the superficial laminae where these axons terminate, it would be important to check, and could be done easily with immunohistochemistry for CGRP, IB4, NF200 combined with mCherry (as was performed in the DRG, fig 2).

5. Figure 4: Again, more detail should be provided, on the pattern of cFos activity, which lamina were cFos+ cells localised, what was the rostro-caudal pattern over C5-C8, and how did this pattern change over time? Only one image is shown, at the 1 month time point (but not specified if this is from the chemogenetic stimulated group or control mCherry). The schematics also do not show the full picture – why only show the control group from a 1 month time point and the chemogenetic stimulated group at the 3 month time point? It is not clear whether the pattern of cFos expression shown in the 2 schematics represents a sum of all cFos+/NeuN+ activated cells across C5-C8 or at one level. Also, "not all c-Fos+ nuclei were in NeuN+" – what are these cells? A detailed reconstruction and lamina distribution of the cFos data would strengthen the manuscript and would provide compelling evidence to support the claim "...neuronal activation not only enhances axon regeneration into spinal gray matter but that these regenerated axons synapse upon interneurons and integrate into spinal circuits."

6. Figures 6 and 7: Data for axon crossing (Fig. 6, panel H) do not seem to replicate the data in figure 1 – since CNO-activated hM3Dq+ neurons were better able to cross the inhibitory substrate than control (mCherry+) neurons, whereas in figure 1 ChABC was required to see any increase in axon crossing in the CSPG spot assay. Also, there seems an interesting pattern between acetylated and tyrosinated tubulin – it would be interesting to look at these in the same axon since the images in figures 6D and 7D seem like the two diverge at the mid-point of the axon, with acetylated tubulin localised proximally and tyrosinated tubulin localised distally. A comparison of this pattern in both hM3Dq+ neurons and control (mCherry+) neurons would be informative. In the experiments presented in figures 6 and 7 (which the authors could consider combining), were the axons cultured in the presence of ChABC? Have comparisons been made with +/- ChABC conditions? This seems important since all previous effects on in vitro and in vivo growth required the presence of ChABC. Finally, what are the effects of ChABC treatment on microtubule dynamics? Have the authors looked - this seems important?

7. Discussion points: The number of regenerating axons, as presented, are few and may not fully account for the impressive grid walk recovery. Discuss the possibility that ChABC treatment is having additional effects on connectivity and neuroplasticity within the spinal cord that is independent of the sensory fibre ingrowth into the dorsal horn. Already enhanced plasticity due to ChABC may have made the spinal cord more amenable to chemogenetic stimulation.

Also, discuss issues of timing – it is interesting that no recovery was seen at early time points (when some active enzyme would still be present) but there was a steady improvement in the weeks after enzyme activity would be diminished. Could the authors speculate on whether it is important to have early and transient neuroplasticity (ChABC) followed by repeated chemogenetic stimulation, or would further improvements likely be observed with continuous ChABC delivery (e.g. with a gene therapy) alongside chemogenetic stimulation?

Minor:

- Fig 2: results p18 “some” cgrp and ib4 neurons were transduced, the vast majority were nf200 – give actual percentages.
- A number of reviews cited in the introduction are dated – some recent topical reviews from the field could be added e.g. Hutson and Di Giovanni (Nat Rev Neurol 2019 15:732-745), Bradbury and Burnside (Nat Commun 2019 10:3879), Griffin and Bradke (EMBO Mol Med 2020 12:e11505).
- There is no evidence that regrowing axons made “90 turns” to enter the dorsal horn (discussion P33) and this speculation should be removed.

Reviewer #2:

Remarks to the Author:

The central claim by Wu et al. is that prolonged elevation of neural activity, achieved through chemogenetics, can enhance axon growth in injured sensory neurons. This claim is supported by in vitro assays in which chemogenetic stimulation increased the ability of axons to cross boundaries of growth-inhibitory aggrecan, and in vivo findings that stimulation increased the invasion of regenerating sensory axons into spinal grey matter. In support of the latter findings, the authors show that stimulation of sensory fibers elevates activity in spinal neurons, indicating functional synapses, and also show partial recovery in a behavioral task. Finally, the authors show in vitro that chemogenetic stimulation alters marks associated with stable and labile microtubules, consistent with a role for microtubule stability in mediating the pro-growth effects of stimulation.

The topic is potentially of high interest, and chemogenetics offers a powerful means to probe the link between activity and growth. The experiments are well designed and the manuscript is well written. One key piece of information is missing, however, which detracts from the potential impact: neuronal activity itself is not measured. Prior work has already established a strong link between neural activity and process outgrowth in various cell types, including the sensory neurons examined here. Thus, while the topic remains highly important, the key questions in the field have moved on to the underlying mechanisms, and the need to resolve conflicting claims and models about the size and even valence of the effect. It is likely that different patterns of activity impact growth quite differently. At the most basic level, the authors need to confirm that the DREADD/CNO approach is succeeding in increasing neural firing, and in the in vivo experiments they should determine how long this effect lasts (e.g. after the single injection are animals being stimulated for two hours? Six hours?). Beyond simply verifying the success of stimulation, the field needs data points to link specific patterns (frequencies) of activity to specific growth phenotypes. The current dataset could be made quite powerful if the evoked patterns of neural activity could be monitored.

Alternatively, regarding mechanistic questions, the findings that activity impinges on microtubule stability is quite interesting but also somewhat incomplete. Specifically, there is no mechanistic link made between the two observations, leaving unaddressed the critical question of how neural activity signals to microtubule modification. Filling in this gap would be another route to a highly impactful study.

In summary, although the work is well done and the phenomenon is important, a more substantial advance would be achieved by either determining the evoked activity and link its pattern to the

evoked growth, or by better elucidating the cellular mechanisms that link activity to microtubule stability.

We sincerely thank the Reviewers for their thoughtful review and comments. We are very pleased that both Reviewers felt that our findings are important. Each of the Reviewers' comments is provided below along with a detailed response. Revisions to the manuscript in response to the comments are noted in the body of the manuscript in blue font.

Reviewers' comments:

Reviewer #1 (Remarks to the Author):

The paper by Tom and co-authors examines sensory axon regeneration into the spinal cord of adult rats using a novel combination approach aimed at stimulating new growth and functional connectivity in the spinal dorsal horn. They used a combination of repeated chemogenetic activation of adult sensory neurons in the dorsal root ganglia, DRG, (by in vivo AAV transduction of the excitatory DREADD receptor hM3Dq and activation with daily CNO injections) and enzymatic modulation of the inhibitory environment at the dorsal root entry zone, DREZ (by intraspinal injection of the neuroplasticity-promoting agent chondroitinase ABC, ChABC). They present evidence of anatomical regrowth, functional connectivity and recovery of in a grid walk task with long term (12 weeks) but not short term (4 weeks) treatment. These are novel and important findings. The grid walk data is particularly convincing (although may not be related only to dorsal horn reinnervation). Some important control groups were not included, so in the absence of these a more in-depth analysis of axonal growth would be appreciated to more fully convince the observed effects are due to the specific treatments. Also, important methodological detail is missing and needs clarification.

1. What happens to the collaterals of the large diameter projections that ascend in the dorsal columns and are important for proprioception? As this population is labelled with mCherry the authors should look at this projection in more rostral segments and, importantly, in their terminal fields in the brainstem dorsal column nuclei. Previous studies using a similar injury (C5-C8 dorsal root crush), treatment paradigm (AAV DRG delivery) and long term (12 week) time course showed regrowth up to the C1 spinal cord and medulla (e.g. Cheah et al 2016 J Neurosci 36, 7286; this paper should be cited); others have shown long-distance ascending sensory axon growth to the medulla but no innervation of the nuclei (e.g. Tuszynski work). It seems important that this projection, not just dorsal horn innervation, is examined.

The mCherry tag that we used to identify neurons that were transduced is fused to hM3Dq and was not expressed at detectable levels throughout the length of the axon. Therefore, due to technical limitations, we cannot assess long axon extension in the dorsal columns, including whether these axons reached the dorsal column nuclei. However, this does not diminish the impact of our findings since the functional recovery we observed may not be dependent upon long-distance regeneration. Afferent

projections into the dorsal horn, particularly onto neurons in lamina III, where we saw evidence of transsynaptic activation, provide sensory feedback that is important for motor function (Koch et al., 2017). We now discuss this more thoroughly.

2. Intraganglionic AAV injections involve direct injections into the C6-C8 DRG (one microliter injected with a glass micropipette, then left in place for 1-2 min before removal). This is performed 2 weeks prior to the dorsal root crush injury, so in effect is likely to cause a “conditioning injury”. A control group should have been included which has no AAV injection, to see how much the injection procedure itself affects DRG growth, and then make the comparison +/- DREADD, to see how much this is amplified by DREADD activation. In absence of this, the authors could examine the DRG (immunohistochemically or biochemically) for injury and growth-associated markers and/or expression of activated pathways involved in neuronal activation-mediated growth e.g. mTOR, JAK/STAT, tubulin-modifying proteins, ATF3, GAP-43). Some analysis of the DRG cell body response across the different treatment groups would significantly strengthen the manuscript.

We agree that intraganglionic injections could cause trauma and activate a “conditioning injury” response. In fact, we have observed this in the past (Wu et al., 2016). However, this alone cannot explain the effect as both the control, mCherry-only controls and the hM3Dq group were similarly injected. Moreover, because we used the mCherry reporter to identify axons from transduced neurons, the suggested addition group of no AAV injection would not be possible or appropriate. While it would be possible to label axons with cholera toxin B injections into the peripheral nerves, this is so unlike what was done with the other groups and would preclude its appropriateness as a control.

The point that the reviewer brought up about a difference in cell body response across the different treatment groups is an excellent one. We include new data that DREADD-mediated activation increases mTOR activation in the DRG cell bodies (Figure 9), further elucidating mechanism underlying the positive effects on growth. We also discuss how mTOR is known to regulate labile microtubules in a way that is consistent with our findings.

3. Methodology detail:

- Study design, blinding and randomisation etc not specified.

We apologize that these details were not clearer. We now indicate that all animals and neuronal cultures used in these studies were randomly assigned to groups prior to treatment or any experimental manipulation. We also indicate that all analyses were performed while blinded to treatment group.

- Frustratingly no n numbers are provided – how many animals were perfused at 4 weeks and 12 weeks post-injury? How many animals/group were used for terminal electrical stimulation studies (to induce cFos) at the 4 week and 12 week time points? In the experimental design section (P14) only minimal information is provided e.g. “Sample sizes were determined based upon our previous published studies using similar techniques”, but then no sample size numbers are actually provided.

We apologize for frustrating the Reviewer. We had stated all of the sample sizes in the relevant figure legends but, to improve clarity, now include them in the methods section, as well.

- CNO administration – add more detail, how many consecutive days did animals receive CNO, what was the total amount injected? State early in the manuscript (e.g the abstract) the time frame and treatment regime e.g. daily CNO injections over 12 weeks post-rhizotomy?

As described in the manuscript, all animals (both hM3Dq and control, mCherry-only) received daily injections of CNO (1mg/kg) over the duration of the experiment, i.e., up to 12 weeks post-dorsal root crush. This is now stated earlier in the manuscript and repeated in multiple places in the text.

- No mention of any adverse effects of daily CNO injections over 12 weeks. This must be included.

We now note in the results section that we did not observe any obvious adverse effects of the daily CNO injections over the 12 weeks.

- More control groups would have strengthened the data i.e. a no-ChABC group; a no-CNO group. Justify why these groups were not included and add some discussion on interpretation of the results in absence of these groups.

The *in vitro*, “screening” studies described in Figure 1 formed the basis for which groups to include in the *in vivo* experiments. The no-ChABC and no-CNO control groups were included in the *in vitro* studies. We found that neuronal activation alone (i.e., in the absence of ChABC) was insufficient to promote axon crossing on CSPG-containing inhibitory boundaries. However, combining neuronal activation with ChABC produced synergistic effects on axon crossing of the rim. Thus, we injected ChABC into all of the animals used for the *in vivo* studies. Likewise, we found that we needed to activate hM3Dq⁺ neurons with CNO to promote axon growth. Expressing hM3Dq but in the absence of CNO (just DMSO vehicle) had no effect on axon growth across the rim. For

the *in vivo* studies, we injected CNO into both hM3Dq⁺ and mCherry⁺ controls to account for any off-target effects CNO may have (MacLaren et al., 2016; Manvich et al., 2018). These are now better described in the text.

- A timeline figure should be included, to clearly show the timings of AAV administration, CNO dosing, rhizotomy, ChABC injection, electrical stimulation and behavioural testing time points.

This is an excellent suggestion. We now include a timeline figure (new Figure 1) to schematize the experimental design.

- P13 “mCherry⁺ axons...were counted using ImageJ” – again, no detail provided - how many sections/animal, what regions were analysed i.e. only the DREZ transition zone, superficial dorsal horn, deep dorsal horn (which lamina?), how many ROI and how were ROI defined?

We apologize that this was not described well. We used methodology that is established in the lab and that we used in similar studies to examine sensory axon regeneration after a dorsal root crush (Wu et al., 2016). For the quantification, we cut the spinal cords with roots from C6 to C7. We collected the sections in a series of 6. We used 6 sections that contained at least a portion of the dorsal root from each animal for quantification. Sections were immunostained with antibodies against mCherry to visualize regenerated axons from AAV5-hSyn-mCherry or -hM3Dq transduced DRG neurons and p75 to visualize Schwann cells within the dorsal root. The DREZ was identified as the interface between p75⁺ dorsal root and p75⁻ spinal cord. Z-stack confocal microscope images presented with maximum intensity projection were used for quantification. While blinded to treatment group, mCherry⁺ axons that extended distal to the p75-defined DREZ and into gray matter were counted using ImageJ and were binned into 50 μ m intervals.

- How was the “completeness” of injury confirmed? There is always some doubt with a crush injury that there will be some spared axons. Clasping of the paw does not confirm completeness, this can only be confirmed anatomically e.g. by tracing and absence of terminal labelling in the cuneate nuclei.

We agree with the Reviewer that a crush injury may spare some axons. This is why we carefully repeated the dorsal root crush procedure 3 times. In our experience validating this method to produce complete dorsal root crush injuries of C5-T1, we very rarely observed animals with incomplete injuries where axons were found across the DREZ. Moreover, those rare animals had residual forepaw function that was in contrast from the typical clasping of the forepaw observed in animals with complete dorsal root crushes. Any animals suspected to have an incomplete injury are always excluded from

analysis. We now provide these details in the Methods. Moreover, the time frame of the sensorimotor recovery observed (6 weeks after crush) is consistent with axon regrowth and not sparing. If there was significant sparing of fibers, and thus sparing of their synaptic connections, the “recovery” would have been observed well before 6 weeks. This is now added to the Discussion.

4. Figure 3 (now Figure 5): Provide a more detailed/comprehensive picture of the extent of regeneration. Multiple dorsal roots were crushed (from C5 to T1), yet only two images are shown and it is not clear which segmental level they are from, or whether the pattern/extent of axon regeneration differs at different levels? Are the counts reflective of the overall number across the entire extent of C5-T1? A rostro-caudal reconstruction across C5-T1 would be informative.

We now indicate in the Methods more clearly that we examined axon regeneration from C6-C7. Because the sections were not collected serially, we cannot provide a rostro-caudal reconstruction across C5-T1. However, we do not feel that this diminishes the significance of our findings, including the identification of a novel mechanism of activation-mediated axon regeneration.

Also, more detail on which fibre type is growing across the DREZ would be informative – they are presumably mostly the large-diameter NF200-positive axons given that a higher percentage of these were transduced than C fibres, however given that ~30% transduced neurons were nociceptive neurons and that cFos expression in fig 4 looks to be more localised in the superficial laminae where these axons terminate, it would be important to check, and could be done easily with immunohistochemistry for CGRP, IB4, NF200 combined with mCherry (as was performed in the DRG, fig 2).

The Reviewer brings up a very good suggestion. Unfortunately, we do not have any more tissue from the regions analyzed in the other figures, preventing us from doing additional stains on that tissue. However, it should be noted that even if some nociceptive afferents regenerated, which is likely, no difference in pain-like sensation was noted between the groups (Figure 4).

5. Figure 4 (now Figure 6): Again, more detail should be provided, on the pattern of cFos activity, which lamina were cFos+ cells localised, what was the rostro-caudal pattern over C5-C8, and how did this pattern change over time? Only one image is shown, at the 1 month time point (but not specified if this is from the chemogenetic stimulated group or control mCherry). The schematics also do not show the full picture – why only show the control group from a 1 month time point and the chemogenetic stimulated group at the 3 month time point? It is not clear whether the pattern of cFos expression shown in the 2 schematics represents a sum of all cFos+/NeuN+ activated

cells across C5-C8 or at one level. Also, “not all c-Fos+ nuclei were in NeuN+” – what are these cells? A detailed reconstruction and lamina distribution of the cFos data would strengthen the manuscript and would provide compelling evidence to support the claim “...neuronal activation not only enhances axon regeneration into spinal gray matter but that these regenerated axons synapse upon interneurons and integrate into spinal circuits.”

We apologize for the confusion. The representative images shown are from both mCherry+ and hM3Dq+ animals at 12 weeks. We show in Figure 6D that the cFos+ cell pattern does indeed change over time, as there are far fewer cells at 4 weeks than at 12 weeks and that there is a treatment-effect. We also provide new data in Figure 6F that cFos+ neurons are present in superficial laminae I-II and lamina III. Notably, interneurons within lamina III are known to be important for integrating sensory feedback on motor function. We only analyzed data from spinal levels C6-C7, the same spinal levels in which we examined axon regeneration. The c-Fos data we obtained are consistent with the axon regeneration data. Thus, we do not believe that not having the rostro-caudal pattern of neuronal activation diminishes the significance of our findings.

The Reviewer brings up an excellent point about the non-neuronal c-Fos+ cells. We now include more analysis of these cells in Supplemental Figure 1 and provide evidence that they include GFAP+ astrocytes.

6. Figures 6 and 7 (now Figures 7 and 8): Data for axon crossing (Fig. 6 [now Fig. 7], panel H) do not seem to replicate the data in figure 1 – since CNO-activated hM3Dq+ neurons were better able to cross the inhibitory substrate than control (mCherry+) neurons, whereas in figure 1 ChABC was required to see any increase in axon crossing in the CSPG spot assay.

We apologize for the confusion. The data in now Fig 7H do in fact replicate the data in Figure 1. In both figures, CNO-activated hM3Dq+ neurons are better able to cross the inhibitory substrate ONLY when the substrate is digested with ChABC. We had noted that ChABC was used in the legend for Fig. 7 but we now clarify this more in the text and the figure itself.

Also, there seems an interesting pattern between acetylated and tyrosinated tubulin – it would be interesting to look at these in the same axon since the images in figures 6D and 7D (now Figures 7D and 8D) seem like the two diverge at the mid-point of the axon, with acetylated tubulin localised proximally and tyrosinated tubulin localised distally. A comparison of this pattern in both hM3Dq+ neurons and control (mCherry+) neurons would be informative. In the experiments presented in figures 6 and 7 (which the authors could consider combining), were the axons cultured in the presence of ChABC? Have comparisons been made with +/- ChABC conditions? This seems important since

all previous effects on in vitro and in vivo growth required the presence of ChABC. Finally, what are the effects of ChABC treatment on microtubule dynamics? Have the authors looked - this seems important?

We agree that the differential pattern of acetylated and tyrosinated tubulin in extending axons from chemogenetically-activated and control neurons is quite interesting. In fact, based on these findings and the followup experiments, we believe that the shift towards a more dynamic microtubule array underlies the improved growth with neuronal activation. Unfortunately, the acetylated and tyrosinated tubulin antibodies we use are raised in the same species, limiting our ability to use them concurrently in the same sample. For this reason and to improve clarity, we would prefer to keep the panels as two separate figures. However, this does not diminish the findings that, *in toto*, indicate that neuronal activation results in a more dynamic microtubule array at the end of growing axons.

For the cultures used for microtubule state assaying, we did not use a CSPG substrate or ChABC. This would be interesting to do in future studies. Unfortunately, the *in vitro* CSPG-ChABC model we used in Figure 1 is good for screening experimental manipulations that allow for crossing of injury-induced inhibitory boundaries but is too high a cell density to allow for easily following individual axons, which is needed here. For these purposes, we chose to use a condition where we could follow individual axons that were actively growing, similar to what we surmise is occurring *in vivo*, to best determine if and how neuronal activation affects microtubule dynamics.

7. Discussion points: The number of regenerating axons, as presented, are few and may not fully account for the impressive grid walk recovery. Discuss the possibility that ChABC treatment is having additional effects on connectivity and neuroplasticity within the spinal cord that is independent of the sensory fibre ingrowth into the dorsal horn. Already enhanced plasticity due to ChABC may have made the spinal cord more amenable to chemogenetic stimulation.

We absolutely agree that ChABC promotes neuroplasticity beyond axon regrowth. We now discuss this possibility in more depth.

Also, discuss issues of timing – it is interesting that no recovery was seen at early time points (when some active enzyme would still be present) but there was a steady improvement in the weeks after enzyme activity would be diminished. Could the authors speculate on whether it is important to have early and transient neuroplasticity (ChABC) followed by repeated chemogenetic stimulation, or would further improvements likely be observed with continuous ChABC delivery (e.g. with a gene therapy) alongside chemogenetic stimulation?

The reviewer brings up some excellent points and we now discuss the timing of ChABC and chemogenetic stimulation. We also discuss that activity plays an important role in the formation of synapses within developing circuits and that that may also play a role in the present studies.

Minor:

- Fig 2 (now Figure 3): results p18 “some” cgrp and ib4 neurons were transduced, the vast majority were nf200 – give actual percentages.

We appreciate the Reviewer’s comments and now state the actual mean percentages \pm SEM in the results.

- A number of reviews cited in the introduction are dated – some recent topical reviews from the field could be added e.g. Hutson and Di Giovanni (Nat Rev Neurol 2019 15:732-745), Bradbury and Burnside (Nat Commun 2019 10:3879), Griffin and Bradke (EMBO Mol Med 2020 12:e11505).

We added these references to the text where relevant.

- There is no evidence that regrowing axons made “90 turns” to enter the dorsal horn (discussion P33) and this speculation should be removed.

The Reviewer’s point is well-taken and that text has been removed.

Reviewer #2 (Remarks to the Author): The central claim by Wu et al. is that prolonged elevation of neural activity, achieved through chemogenetics, can enhance axon growth in injured sensory neurons. This claim is supported by in vitro assays in which chemogenetic stimulation increased the ability of axons to cross boundaries of growth-inhibitory aggrecan, and in vivo findings that stimulation increased the invasion of regenerating sensory axons into spinal grey matter. In support of the latter findings, the authors show that stimulation of sensory fibers elevates activity in spinal neurons, indicating functional synapses, and also show partial recovery in a behavioral task. Finally, the authors show in vitro that chemogenetic stimulation alters marks associated with stable and labile microtubules, consistent with a role for microtubule stability in mediating the pro-growth effects of stimulation.

The topic is potentially of high interest, and chemogenetics offers a powerful means to probe the link between activity and growth. The experiments are well designed and the manuscript is well written. One key piece of information is missing, however, which detracts from the potential impact: neuronal activity itself is not measured. Prior work

has already established a strong link between neural activity and process outgrowth in various cell types, including the sensory neurons examined here. Thus, while the topic remains highly important, the key questions in the field have moved on to the underlying mechanisms, and the need to resolve conflicting claims and models about the size and even valence of the effect. It is likely that different patterns of activity impact growth quite differently. At the most basic level, the authors need to confirm that the DREADD/CNO approach is succeeding in increasing neural firing, and in the in vivo experiments they should determine how long this effect lasts (e.g. after the single injection are animals being stimulated for two hours? Six hours?). Beyond simply verifying the success of stimulation, the field needs data points to link specific patterns (frequencies) of activity to specific growth phenotypes. The current dataset could be made quite powerful if the evoked patterns of neural activity could be monitored.

Alternatively, regarding mechanistic questions, the findings that activity impinges on microtubule stability is quite interesting but also somewhat incomplete. Specifically, there is no mechanistic link made between the two observations, leaving unaddressed the critical question of how neural activity signals to microtubule modification. Filling in this gap would be another route to a highly impactful study.

In summary, although the work is well done and the phenomenon is important, a more substantial advance would be achieved by either determining the evoked activity and link its pattern to the evoked growth, or by better elucidating the cellular mechanisms that link activity to microtubule stability.

We appreciate the Reviewer's thoughtful and positive comments. It would indeed be interesting to determine how a different patterns of neuronal firing affect axon regeneration. However, the nature of chemogenetics does not allow for such precise control over neuronal firing patterning. Optogenetics or electrical stimulation would be a better means to address this important question, given that it is easier to more tightly regulate frequency and duration of neuronal activation with these techniques. This is now discussed.

The Reviewer also brought up that it is important to understand mechanism underlying microtubule modification triggered by neural activity. This was also brought up by Reviewer 1. We provide new data indicating that chemogenetically-activated DRG neurons have more mTOR activation (indicated by p-S6 expression) than control, mCherry⁺ neurons. mTOR is known to foster dynamic microtubules, as we see here. We discuss this in more detail in the Discussion.

Importantly, we feel that a significant aspect of this study is the identification of a mechanism (i.e., shifting of the microtubule array in the distal axon towards being more dynamic) that we directly show mediates neuronal activation-mediated axon growth. Moreover, it should be stressed that while many studies focus on upstream mechanisms, such as mTOR, it is also important to identify more downstream

mechanisms that underlie improved axon regeneration. Additionally, considering there has been a lot of recent focus in the field on stabilizing microtubules to enhance axon regeneration, it is our belief that our findings demonstrating that swinging the distal axonal microtubule array towards being more dynamic - and less stable – fosters functional axon regeneration do advance the field substantially.

Reviewers' Comments:

Reviewer #1:

Remarks to the Author:

The authors have done an excellent job with the revised manuscript. The additional data is compelling, particularly the dorsal root ganglia analysis. The text additions have significantly enhanced the manuscript and provide strong support for their conclusions. This is an excellent paper and an important addition to our field.

Reviewer #2:

Remarks to the Author:

In the prior round of review, the central point was that although the work was of potential interest, two significant gaps existed. The first is that neuronal activity, the titular focus of the study, was not verified or measured. Especially in the *in vivo* work, we simply don't know how much activity increased, or for how long – or indeed whether it increased at all. At the most fundamental level, if an experiment is conducted without monitoring the independent variable, how are the results to be interpreted?

On this point the authors have been unresponsive, other than to point out that chemogenetics offers less control over activity than optogenetics. The reviewer is aware of this distinction. This response misconstrues the prior review. The request was not to achieve tight control over the independent variable, that is, use optogenetics to systematically vary patterns of activity and measure growth. Although such an experiment would be highly informative, the demand would be unreasonable in review. Rather, the prior review was a (more reasonable) request to simply monitor the independent variable, that is, to supply chemogenetic stimulation as described and then provide information about the magnitude and duration of activation. The manuscript is unchanged with regard to this point, and the fundamental concern remains.

The second main point regarded the proposed mechanism by which activity regulates axon growth. The authors indicate a role for microtubule stability. In general terms, the notion that axon growth is affected by microtubule stability is long-standing in the field of axon growth and guidance. More recently it has been explored in the context of sensory axon regeneration by the work of Dr. Frank Bradke. Overall it is expected that an extending axon would require an appropriate level of microtubule stability.

The potential advance would be to clarify the mechanism by which neuronal activity influences microtubule stability in the axon. Here there has been some movement. The authors now supply evidence that chemogenetic activation increases mTOR activity in sensory neurons, and then speculate on potential links between mTOR and microtubule stability. It is important to note that prior work has already established the link between neural activation and mTOR activity (Zareen et al. 2018). Although a step in the right direction, the findings remain correlative and speculative, and do not provide substantial or novel mechanistic insight.

In summary, the overall enthusiasm of this review is only slightly shifted from the prior round, in line with the marginal changes in the manuscript itself. The field is well aware that neural activity influences axon growth, and that microtubule stability is an important downstream effector for axon growth in general. As it stands, the manuscript is an exemplar of these understandings; useful and well done, but not a major advance. What the field doesn't understand is the (likely complex) correlation between different patterns of neural activity and growth effects, nor the intracellular mechanisms that link activity to growth (although prior work does hint at mTOR and Jak/Stat involvement). As the prior review pointed out, there is an opportunity here to address those gaps. The manuscript revisions, unfortunately, have moved only fractionally in that direction.

We again thank the Reviewers for their thoughtful review and comments. Each of the Reviewers' comments is provided below along with a detailed response. Revisions to the manuscript in response to the comments continue to be noted in the body of the manuscript in blue font.

REVIEWER COMMENTS

Reviewer #1 (Remarks to the Author):

The authors have done an excellent job with the revised manuscript. The additional data is compelling, particularly the dorsal root ganglia analysis. The text additions have significantly enhanced the manuscript and provide strong support for their conclusions. This is an excellent paper and an important addition to our field.

We thank the Reviewer for their thoughtful comments during the review process that resulted in an improved manuscript and their enthusiastic endorsement of this manuscript.

Reviewer #2 (Remarks to the Author):

In the prior round of review, the central point was that although the work was of potential interest, two significant gaps existed. The first is that neuronal activity, the titular focus of the study, was not verified or measured. Especially in the *in vivo* work, we simply don't know how much activity increased, or for how long – or indeed whether it increased at all. At the most fundamental level, if an experiment is conducted without monitoring the independent variable, how are the results to be interpreted?

On this point the authors have been unresponsive, other than to point out that chemogenetics offers less control over activity than optogenetics. The reviewer is aware of this distinction. This response misconstrues the prior review. The request was not to achieve tight control over the independent variable, that is, use optogenetics to systematically vary patterns of activity and measure growth. Although such an experiment would be highly informative, the demand would be unreasonable in review. Rather, the prior review was a (more reasonable) request to simply monitor the independent variable, that is, to supply chemogenetic stimulation as described and then provide information about the magnitude and duration of activation. The manuscript is unchanged with regard to this point, and the fundamental concern remains.

We apologize for the crossed channels. We agree that it is important to confirm the independent variable of chemogenetic activation. We performed new experiments to verify *in vivo* activation of DRG neurons transduced to express hM3Dq. We found that injecting CNO into animals results in more c-Fos expression, an established means to

verify chemogenetic activation, in hM3Dq⁺ DRGs than in control, mCherry⁺ DRGs (see new Figure 3).

We agree with the Reviewer that it would be highly interesting to provide information about the magnitude and duration of activation. After conferring with several colleagues with expertise in electrophysiology, including with recording from dorsal root ganglion neurons *in vitro*, we came to the conclusion that it would be virtually impossible to record from individual dorsal root ganglion neurons *in vivo*. The closest we could get *in vivo* would be to perform dorsal root recordings, which is not trivial and would take a considerable amount of time (perhaps upward of a year) and effort to successfully and reliably perform them using completely new equipment setups. It is more feasible to record from chemogenetically-activated dorsal root ganglion neurons *in vitro*, but we would not be able to discern duration of activation *in vivo*, as specifically noted by the Reviewer. While it would be interesting to have these data, we do not believe that they are necessary to support our conclusions that chronic chemogenetic activation increases functional sensory axon regeneration via dynamic microtubules. Moreover, we are not making any claims that stimulation of a particular magnitude nor duration is needed to observe the effects here. As such, we do not feel that the absence of these data diminish the importance or novelty of our findings.

The second main point regarded the proposed mechanism by which activity regulates axon growth. The authors indicate a role for microtubule stability. In general terms, the notion that axon growth is affected by microtubule stability is long-standing in the field of axon growth and guidance. More recently it has been explored in the context of sensory axon regeneration by the work of Dr. Frank Bradke. Overall it is expected that an extending axon would require an appropriate level of microtubule stability.

The potential advance would be to clarify the mechanism by which neuronal activity influences microtubule stability in the axon. Here there has been some movement. The authors now supply evidence that chemogenetic activation increases mTOR activity in sensory neurons, and then speculate on potential links between mTOR and microtubule stability. It is important to note that prior work has already established the link between neural activation and mTOR activity (Zareen et al. 2018). Although a step in the right direction, the findings remain correlative and speculative, and do not provide substantial or novel mechanistic insight.

In summary, the overall enthusiasm of this review is only slightly shifted from the prior round, in line with the marginal changes in the manuscript itself. The field is well aware that neural activity influences axon growth, and that microtubule stability is an important downstream effector for axon growth in general. As it stands, the manuscript is an exemplar of these understandings; useful and well done, but not a major advance. What the field doesn't understand is the (likely complex) correlation between different patterns of neural activity and growth effects, nor the intracellular mechanisms that link activity to growth (although prior work does hint at mTOR and Jak/Stat involvement). As the prior

review pointed out, there is an opportunity here to address those gaps. The manuscript revisions, unfortunately, have moved only fractionally in that direction.

It appears that the Reviewer misunderstood what we feel is the ultimate (and novel) mechanism of chemogenetic activation-mediated growth, i.e., shifting the microtubule array in an extending axon towards being more dynamic. The Reviewer stated that microtubule stability has already been explored in the context of sensory regeneration by Frank Bradke. We agree that Dr. Bradke and colleagues have explored microtubule stability. However, this was in the context of INCREASING microtubule stability to enhance axon growth. In our studies, we demonstrate that chemogenetic activation enhances growth by shifting the microtubule array towards a more DYNAMIC phenotype, and that post-translational markers of stable microtubules DECREASE in the distal axon. Thus, our findings are, in some ways, the opposite of Dr. Bradke's, which is becoming dogma in the field. More importantly, to our knowledge, this has not been shown before and is, in fact, a novel mechanism for how chemogenetic activity influences axon growth. Given the likely complexity of how activation affects growth, as noted by the Reviewer, we believe that pinpointing a very downstream target - a convergence point for multiple signaling pathways - that directly impacts axon growth is significant. We revised the discussion to make this more clear.

Reviewers' Comments:

Reviewer #2:

Remarks to the Author:

With the new cFos analysis, the authors now provide some validation of neuronal activation via chemogenetic stimulation. Although electrophysiology or calcium imaging would have been more informative, these data provide at least some insight into the amount of activation achieved.

Regarding the novelty of the mechanistic experiments and the insight into microtubule dynamics, perhaps the situation is simply that this reviewer's basic understanding was set decades ago, in the 1990s. Classic experiments that applied compounds to growing axons in culture made it clear that growth depends on the right level of microtubule stability. Dr. Baas is doubtless well aware of this work, and it seems obvious that one can prevent growth both by over-stabilizing or by excessively destabilizing microtubules - and that in principle either mechanism could block growth in vivo. Perhaps, as the authors argue, if a notion that increased microtubule stability is always beneficial is somehow becoming embedded in the field's thinking, it is valuable to provide a counter-example to drive home the point that in fact growth depends on the right balance.

All in all, I appreciate the authors' responsiveness and believe the revised manuscript will be a useful addition to the literature.

Response to the Reviewers' Comments

Reviewer #1: No comments provided for this revision.

Reviewer #2 (Remarks to the Author):

With the new cFos analysis, the authors now provide some validation of neuronal activation via chemogenetic stimulation. Although electrophysiology or calcium imaging would have been more informative, these data provide at least some insight into the amount of activation achieved.

Regarding the novelty of the mechanistic experiments and the insight into microtubule dynamics, perhaps the situation is simply that this reviewer's basic understanding was set decades ago, in the 1990s. Classic experiments that applied compounds to growing axons in culture made it clear that growth depends on the right level of microtubule stability. Dr. Baas is doubtless well aware of this work, and it seems obvious that one can prevent growth both by over-stabilizing or by excessively destabilizing microtubules - and that in principle either mechanism could block growth in vivo. Perhaps, as the authors argue, if a notion that increased microtubule stability is always beneficial is somehow becoming embedded in the field's thinking, it is valuable to provide a counter-example to drive home the point that in fact growth depends on the right balance.

All in all, I appreciate the authors' responsiveness and believe the revised manuscript will be a useful addition to the literature.

We thank the Reviewer for the thoughtful and constructive comments throughout this review process.